# Control of a hippocampal recurrent excitatory circuit by cannabinoid receptor-interacting protein Gap43

Irene B. Maroto [1,2,3], Carlos Costas-Insua [1,2,3], Coralie Berthoux[4], Estefanía Moreno [5], Andrea Ruiz-Calvo[1,2,3], Carlos Montero-Fernández[1], Andrea Macías-Camero [1], Ricardo Martín[1,6], Nuria García-Font[1,6], José Sánchez-Prieto[1,6], Giovanni Marsicano [7], Luigi Bellocchio[7], Enric I. Canela [5], Vicent Casadó [5], Ismael Galve-Roperh [1,2,3], Ángel Núñez [8], David Fernández de Sevilla [8], Ignacio Rodríguez-Crespo[1,2,3], Pablo E. Castillo [4,9] & Manuel Guzmán [1,2,3] ✉

The type-1 cannabinoid receptor (CB₁R) is widely expressed in excitatory and inhibitory nerve terminals, and by suppressing neurotransmitter release, its activation modulates neural circuits and brain function. While the interaction of CB₁R with various intracellular proteins is thought to alter receptor signaling, the identity and role of these proteins are poorly understood. Using a high-throughput proteomic analysis complemented with an array of in vitro and in vivo approaches in the mouse brain, we report that the *C*-terminal, intracellular domain of CB₁R interacts specifically with growth-associated protein of 43 kDa (GAP43). The CB₁R-GAP43 interaction occurs selectively at mossy cell axon boutons, which establish excitatory synapses with dentate granule cells in the hippocampus. This interaction impairs CB₁R-mediated suppression of mossy cell to granule cell transmission, thereby inhibiting cannabinoid-mediated anti-convulsant activity in mice. Thus, GAP43 acts as a synapse type-specific regulatory partner of CB₁R that hampers CB₁R-mediated effects on hippocampal circuit function.

The endocannabinoid system comprises cannabinoid receptors, their lipid ligands (the so-called endocannabinoids), and the enzymatic machinery required for endocannabinoid synthesis, deactivation, and bioconversion[1,2]. The endocannabinoids 2-arachidonoylglycerol and anandamide, as well as the exogenous cannabinoid Δ⁹-tetrahydrocannabinol (THC), the main psychoactive component of cannabis, bind to and activate type-1 and type-2 cannabinoid receptors (CB₁R and CB₂R, respectively), which are evolutionarily-conserved

[1]Department of Biochemistry and Molecular Biology, Instituto Universitario de Investigación Neuroquímica (IUIN), Complutense University, 28040 Madrid, Spain. [2]Centro de Investigación Biomédica en Red sobre Enfermedades Neurodegenerativas (CIBERNED), Instituto de Salud Carlos III, 28029 Madrid, Spain. [3]Instituto Ramón y Cajal de Investigación Sanitaria (IRYCIS), 28034 Madrid, Spain. [4]Dominick P. Purpura Department of Neuroscience, Albert Einstein College of Medicine, Bronx, NY 10461, USA. [5]Department of Biochemistry and Molecular Biomedicine, Faculty of Biology and Institute of Biomedicine of the University of Barcelona, University of Barcelona, 08028 Barcelona, Spain. [6]Instituto de Investigación Sanitaria del Hospital Clínico San Carlos (IdISSC), 28040 Madrid, Spain. [7]Institut National de la Santé et de la Recherche Médicale (INSERM) and University of Bordeaux, NeuroCentre Magendie, Physiopathologie de la Plasticité Neuronale, U1215, 33077 Bordeaux, France. [8]Department of Anatomy, Histology and Neuroscience, School of Medicine, Autónoma University, 28029 Madrid, Spain. [9]Department of Psychiatry and Behavioral Sciences, Albert Einstein College of Medicine, Bronx, NY 10461, USA. ✉e-mail: mguzman@quim.ucm.es

members of the G protein-coupled receptor (GPCR) superfamily[1,2]. CB$_1$R is one of the most abundant GPCRs in the mammalian central nervous system, and its activation mediates retrograde suppression of neurotransmitter release in a short- and long-term manner[3,4]. Thus, CB$_1$R regulates a plethora of body functions, including learning and memory, emotions, feeding and energy metabolism, pain response, and motor behavior[5,6]. Despite the vast number of reports on CB$_1$R-modulated neurobiological processes, studies addressing the precise molecular mechanisms and signaling partners of CB$_1$R at the synapse level remain scarce. CB$_1$R triggers a wide range of downstream cascades that regulate synaptic function and neuronal activity in a markedly context-dependent manner[7]. We and others have previously proposed that the interaction of CB$_1$R with various cytoplasmatic proteins[2,8,9], as well as plasma-membrane GPCRs[10,11], may fine-tune CB$_1$R signaling in vivo. However, the precise functional relevance of these CB$_1$R protein–protein interactions in the brain has not been yet fully elucidated.

CB$_1$Rs are present in both excitatory and inhibitory nerve terminals and their activation can modify the excitatory/inhibitory balance. Studies conducted on conditional CB$_1$R knockout and genetic-rescue mice have revealed that CB$_1$Rs located on excitatory presynaptic boutons, despite their moderate levels of expression compared to GABAergic terminals[3,12,13], act as a synaptic circuit breaker that is crucial for the control of brain excitability[14,15]. Thus, activation of glutamatergic-neuron CB$_1$Rs mediates key (endo)cannabinoid-evoked processes such as hyperphagia[16], anxiolysis[17], neuroprotection[18], and anti-convulsion[19]. Uniquely high levels of glutamatergic-neuron CB$_1$Rs occur in the axon boutons of hilar mossy cells (MCs) of the dentate gyrus (DG)[13,19,20], a region that critically processes information from the entorhinal cortex (EC) to the hippocampal formation. MC boutons are located in the inner molecular layer (IML) and form excitatory synapses with the proximal dendrites of granule cells (GCs), the main excitatory neurons in the DG. In turn, GCs project back to MCs, thereby establishing an associative GC-MC-GC excitatory circuit that gates information transfer from the EC to CA3, and is involved in processing various forms of memory and driving hyperexcitability-evoked epileptic seizures[21,22]. Unlike CB$_1$Rs located on EC-projecting axon boutons, whose activation can even potentiate excitatory synaptic transmission[23], activation of CB$_1$Rs located on MCs suppresses synaptic transmission and hinders long-term potentiation of MC-GC synaptic transmission and GC output[24,25], thus supporting an anti-convulsant action[19].

Here, using a high-throughput proteomic approach complemented with a wide array of in vitro and in vivo assays, we unveil that CB$_1$R interacts specifically with growth-associated protein of 43 kDa (GAP43; aka neuromodulin), a major presynaptic protein that is involved in neurite outgrowth, axonal regeneration, and synaptic plasticity[26]. Moreover, we show that the CB$_1$R-GAP43 interaction is enhanced by GAP43 phosphorylation, occurs selectively in MC axons impinging on GC dendrites, hampers CB$_1$R-mediated depression at MC-GC synapses, and impairs cannabinoid-evoked anti-convulsant activity. Thus, our findings identify GAP43 as a CB$_1$R-interacting protein that regulates receptor function in a synapse-specific manner.

## Results

### Identification of GAP43 as a CB$_1$R-interacting protein

We initially identified GAP43 as a potential CB$_1$R-interacting protein in a high-throughput screening conducted by affinity chromatography and subsequent proteomic analysis. As the large *C*-terminal domain (CTD) encompasses the bulk of the cytoplasmic domain of CB$_1$R, we used recombinant hCB$_1$R-CTD (amino acids 408-472) as bait. A sheep whole-brain homogenate, aimed to allow a large amount of starting biological material, was passed through a lectin-hCB$_1$R-CTD Sepharose 4B column. After washing and elution with lactose, the resulting proteins were digested and subjected to tandem mass spectrometry (Fig. 1a). We obtained a list of ~50 potential CB$_1$R-interacting candidate proteins (Supplementary Table 1). While some of the hits, such as plasma membrane Ca$^{2+}$ ATPases, G-protein α subunits (specifically, Gα$_{i1}$), Na$^+$ and Cl$^-$-dependent GABA transporters, heat shock protein 70, and mitogen-activated protein kinase family members coincided with those found in similar high-throughput studies[27,28], our list also included the pleiotropic protein GAP43/neuromodulin. A Gene ontology (GO) enrichment-based cluster analysis of the list of proteins was performed using the STRING Database and MCL Clustering. We identified two enriched functional GO terms, both including CB$_1$R and GAP43: GO.0008037-Cell recognition (with matching proteins CNR1, CRTAC1, GAP43 and MFGE8), and GO.0008038-Neuronal recognition (with matching proteins CNR1, CRTAC1 and GAP43) ($p = 0.0384$ for each GO term). Based on its presynaptic localization and anatomical distribution in the central nervous system (see below), which raised the possibility of a functional interaction with CB$_1$R, we focused our further analyses on GAP43.

As a first approach to validate a potential CB$_1$R-GAP43 interaction, we used a fluorescence polarization-based binding assay to detect direct protein–protein interactions in vitro[9] (Fig. 1b). We tested a fixed concentration of purified, 5-(iodoacetamido)fluorescein-labeled hCB$_1$R-CTD, and increasing concentrations of purified, unlabeled hGAP43. A saturating polarization curve with $K_d = 38.2 \pm 10.8$ μM ($n = 3$ independent experiments) was obtained, which supports a direct, specific, and high-affinity interaction between GAP43 and CB$_1$R-CTD. To assess the CB$_1$R-GAP43 interaction in neural tissue, we performed co-immunoprecipitation assays in primary mouse hippocampal neurons (Fig. 1c, upper panel) and in mouse whole-hippocampus extracts (Fig. 1c, lower panel). These experiments indicated that endogenous GAP43 and CB$_1$R interact in the mouse brain in vivo. As both GAP43 and CB$_1$R are predominantly located on presynaptic boutons[3,26], we subsequently tested whether they interact at this particular subcellular site using synaptosomal preparations isolated from the mouse hippocampus. Immunostaining of synaptosomes revealed that GAP43 and CB$_1$R were present in 19.7 ± 3.1% (GAP43) or 20.1 ± 4.3% (CB$_1$R) of total synaptophysin 1-positive boutons, and that 8.2 ± 1.8% of total synaptophysin 1-positive boutons were double-positive for GAP43 and CB$_1$R ($n = 5$ independent synaptosomal preparations; Fig. 1d). When restricting these analyses to the GAP43-positive pool of synaptosomes (synaptophysin-1/GAP43 double-positive) or the CB$_1$R-positive pool of synaptosomes (synaptophysin-1/CB$_1$R double-positive), we observed a higher percentage of colocalization between GAP43 and CB$_1$R (within GAP43-positive synaptosomes, 41.6 ± 5.1% were double-positive for GAP43 and CB$_1$R; and within CB$_1$R-positive synaptosomes, 40.4 ± 1.1% were double-positive for GAP43 and CB$_1$R; $n = 5$ independent synaptosomal preparations). These data point to a restricted location of CB$_1$R-GAP43 complexes within the whole hippocampus. Moreover, to evaluate a potential protein–protein interaction in synaptosomes, we performed in situ proximity ligation assay (PLA) experiments, which allow the immunofluorescence-based detection of protein complexes in fixed biological material. Hippocampal synaptosomal preparations from WT mice showed an overt CB$_1$R-GAP43 PLA-positive signal, which was significantly reduced in synaptosomes from CB$_1$R-deficient (*Cnr1*$^{-/-}$) mice (Fig. 1e). Taken together, these observations support a physical interaction between GAP43 and CB$_1$R at a selective pool of mouse hippocampal presynaptic boutons.

### Phosphorylation of GAP43 at S41 facilitates its interaction with CB$_1$R

Previous studies have shown that phosphorylation of GAP43 at S41 by protein kinase C (PKC) is critical for its biological activity[26,29,30]. We thus designed two mutant versions of GAP43 (harboring a phospho-resistant S41A or phospho-mimetic S41D point mutation, respectively) to modulate the activation state of the protein (Fig. 2a). First, PLA was conducted in HEK293T cells co-transfected with a myc-tagged CB$_1$R plus the different forms of GAP43, namely, GFP-GAP43(WT), GFP-GAP43(S41D)

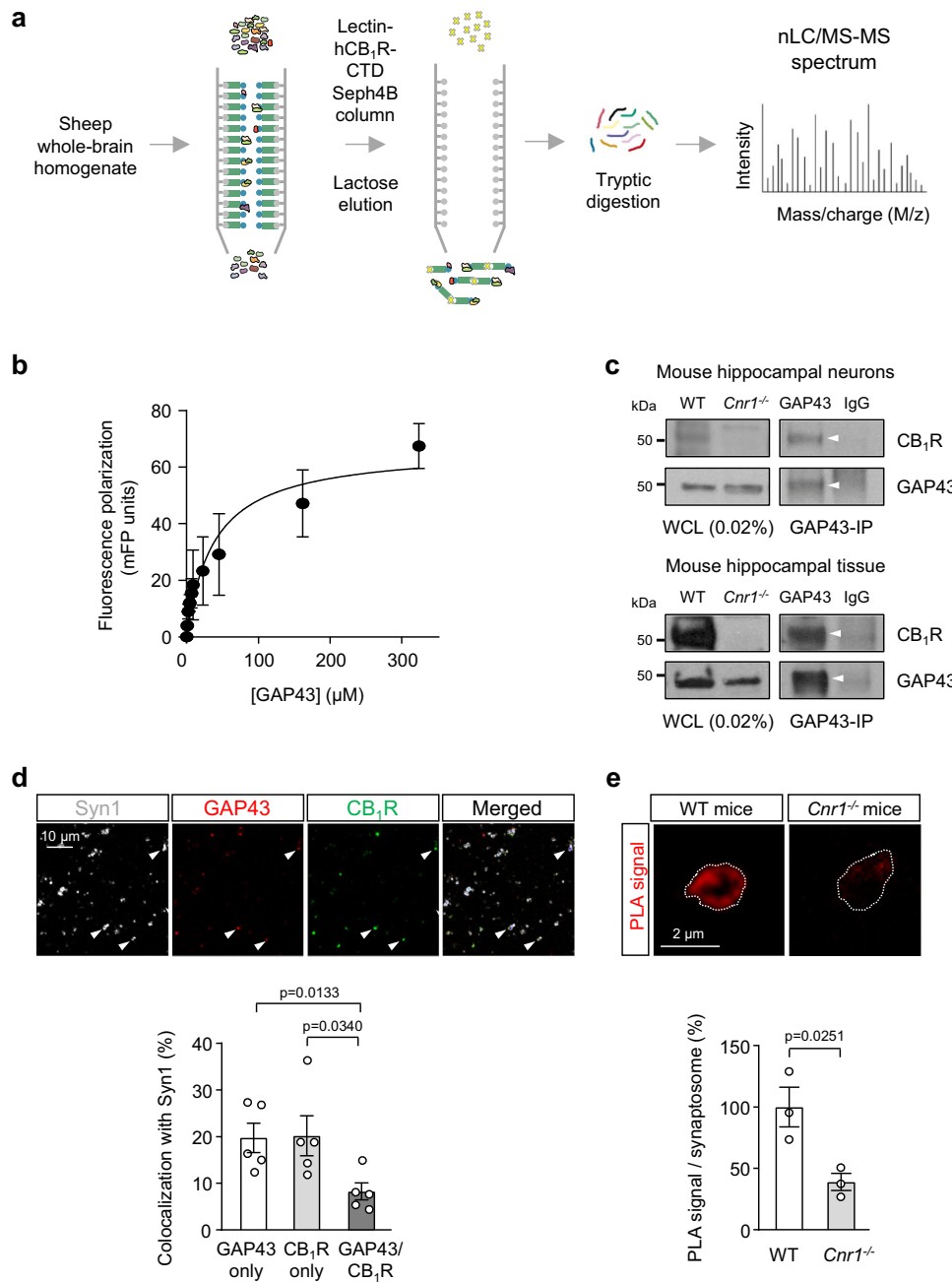

**Fig. 1 | GAP43 interacts with CB1R. a** Schematic workflow of the affinity pur-
ification and tandem MS/MS experiment conducted. A sheep whole-brain homo-
genate was loaded onto a lectin-hCB$_1$R-CTD-bound Sepharose 4B column. After
washing, elution with lactose, eluted-fraction separation by SDS-PAGE, and diges-
tion with trypsin, peptides were subjected to nLC/MS-MS proteomic analysis.
**b** Fluorescence polarization (FP)-based protein–protein binding experiments using
5-IAF-labeled CB$_1$R-CTD and increasing amounts of unlabeled GAP43. FP was
expressed as milli-FP units. Each point represents the mean ± SEM of 3 independent
experiments. **c** Co-immunoprecipitation experiments in (top) primary mouse hip-
pocampal neurons or (bottom) mouse hippocampal tissue. Immunoprecipitation
(IP) was conducted with anti-GAP43 antibody or control IgG. Arrowheads point to
specific precipitated bands. Whole-cell lysates (WCL) from 3-month-old WT and
control *Cnr1*$^{-/-}$ mice are shown. A representative experiment is shown. The

experiment was repeated independently 3 times with similar results. **d** Top,
Representative confocal images of hippocampal synaptosomes of WT mice
immunostained for synaptophysin 1 (Syn1), GAP43 and CB$_1$R. Arrowheads point to
representative triple-colocalizing synaptosomes. Bottom, Quantification of the
percentage of Syn1+ synaptosomes that colocalize with either CB$_1$R only, GAP43
only, or both CB$_1$R and GAP43 (means ± SEM; *n* = 5 independent synaptosomal
preparations; two-tailed unpaired Student's *t* test). **e** PLA for CB$_1$R and GAP43 was
performed in hippocampal synaptosomes from WT mice and *Cnr1*$^{-/-}$ mice as con-
trol. Representative confocal images of CB$_1$R-GAP43 complexes appearing as red
signal (top), and quantification of PLA-positive signal per synaptosome (bottom;
means ± SEM; *n* = 3 independent synaptosomal preparations per genotype; two-
tailed unpaired Student's *t* test). Source data are provided as a Source data file.

or GFP-GAP43(S41A). GAP43(WT)-CB$_1$R and GAP43(S41D)-CB$_1$R com-
plexes were readily detectable and quantified as PLA-positive puncta in
GFP-positive cells, while remarkably lower complex levels were found
in cells transfected with GFP-GAP43(S41A) (Fig. 2b). Second, cells wer-
e co-transfected with HA-CB$_1$R and each of the GAP43 mutants. Upon

HA-CB$_1$R immunoprecipitation using anti-HA antibody and blotting
with an anti-*pan*-GAP43 antibody, GAP43(S41D) was the predominant
co-immunoprecipitated form of the protein (Fig. 2c). Third, biolumi-
nescence resonance energy transfer (BRET) experiments, which allow a
dynamic and highly sensitive detection of protein–protein interactions

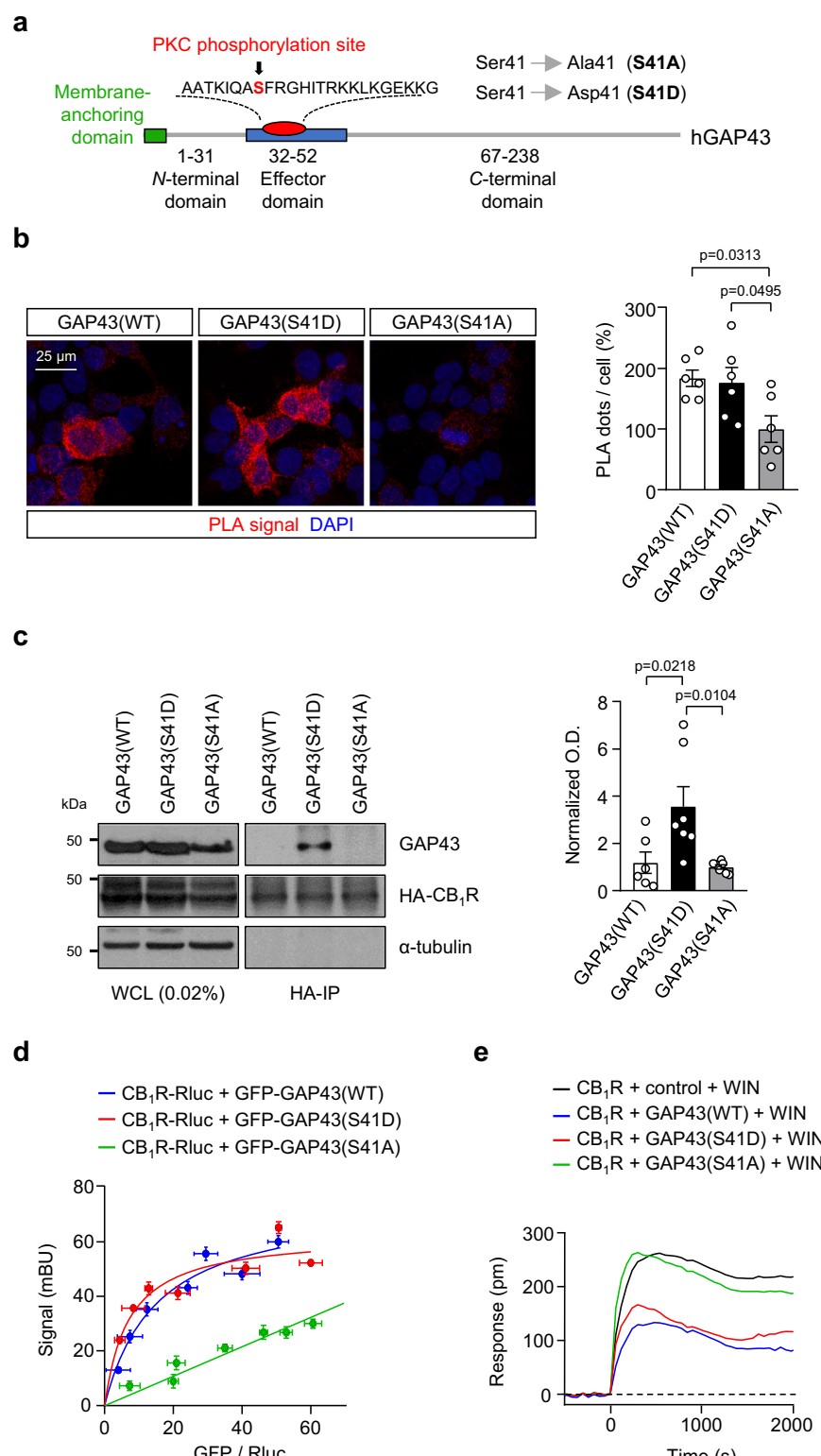

in live cells, were conducted with an RLuc-tagged version of CB$_1$R (Fig. 2d). We found a positive and saturating BRET signal for CB$_1$R-RLuc plus GFP-GAP43(WT) or GFP-GAP43(S41D), indicating a specific protein–protein interaction. Quantitative analysis of the BRET curves supported a higher affinity of CB$_1$R-RLuc for GFP-GAP43(S41D) than for GFP-GAP43(WT) [Fig. 2d; values of RLuc/GFP ratio that give half-maximal BRET effect (BRET$_{50}$), GFP-GAP43(WT): $15.18 \pm 4.53$, $n = 3$ experiments; GFP-GAP43(S41D): $7.08 \pm 2.51$, $n = 3$ independent experiments; $t_{(4)} = 4.216$; $p = 0.0135$ by two-tailed unpaired Student's $t$ test]. In

contrast, the pair CB$_1$R-RLuc/GFP-GAP43(S41A) gave a linear, non-specific BRET signal, indicating an absence of protein–protein interaction ($n = 3$ independent experiments).

We next asked whether GAP43 binding affects CB$_1$R activity. To this end, we performed dynamic mass redistribution (DMR) assays to quantify changes in the overall signaling triggered by agonist-evoked receptor activation (Fig. 2e). We and others have previously used this approach to evaluate CB$_1$R signaling in response to various manipulations[9–11]. When HEK293T cells expressing CB$_1$R were treated

**Fig. 2 | Phosphorylation of GAP43 at S41 facilitates its interaction with CB1R.**
**a** Scheme of the mutant constructs aimed to modify GAP43 activation state. **b** PLA
for CB1R and GAP43 was performed with anti-c-myc and anti-GFP antibodies in
HEK293T cells transfected with CB1R-myc plus GFP-GAP43(WT), GFP-GAP43(S41D)
or GFP-GAP43(S41A). Left, Representative confocal microscopy images show
CB1R-GAP43 complexes appearing as red dots. Cell nuclei were stained with DAPI
(blue). Right, Quantification of PLA-positive dots per GFP-transfected cell. Values of
GFP-GAP43(S41A) were set at 100% (means ± SEM; $n = 6$ independent experiments;
one-way ANOVA with Tukey's multiple comparisons test). **c** Left, Co-
immunoprecipitation experiments in HEK293T cells co-transfected with HA-tagged
CB1R and GAP43(WT), GAP43(S41D), GAP43(S41A). Whole-cell lysates (WCL) are
shown. Right, Quantification of optical density (O.D.) values of co-
immunoprecipitated GAP43 relative to those of HA-CB1R are shown. Values of GFP-
GAP43(S41A) were set at 1 (means ± SEM; GAP43(WT) $n = 6$ independent experi-
ments, GAP43(S41D) $n = 7$ independent experiments, GAP43(S41A) $n = 7$ indepen-
dent experiments; one-way ANOVA with Tukey's multiple comparisons test).
**d** BRET saturation experiments in HEK293T cells expressing CB1R-RLuc and
increasing amounts of GFP-GAP43(WT), GFP-GAP43(S41D) or GFP-GAP43(S41A).
BRET is expressed as milli-BRET units (mBU) (means ± SEM; $n = 3$ independent
experiments). **e** DMR assays in HEK293T cells transfected with CB1R plus
GAP43(WT), GAP43(S41D), GAP43(S41A) or a control empty vector, and exposed to
100 nM WIN-55,212−2 (WIN). A representative experiment is shown ($n = 3$ inde-
pendent experiments). Source data are provided as a Source data file.

with WIN-55,212-2, a widely used synthetic cannabinoid full agonist
that exhibits a high affinity, efficacy, and potency on CB1R[2], we
found that both GAP43(WT) and GAP43(S41D) blunted CB1R action,
conceivably by physically interacting with the receptor, while
this inhibitory effect was not evident with the non CB1R-interacting
mutant GAP43(S41A) [Fig. 2e; values of DMR-signal shift at
peak (DMR$_{max}$), Control: 262.6 ± 8.0 pm, $n = 3$ independent experi-
ments; GAP43(WT): 133.8 ± 18.6 pm, $n = 3$ independent experiments;
GAP43(S41D): 167.9 ± 38.1 pm, $n = 3$ independent experiments;
GAP43(S41A): 263.5 ± 12.4 pm, $n = 3$ independent experiments; $F_{(3, 8)} =$
26.2; Control vs GAP43(WT) $p = 0.0005$ or vs GAP43(S41D) $p = 0.0037$,
and GAP43(S41A) vs GAP43(WT) $p = 0.0005$ or vs GAP43(S41D)
$p = 0.0035$, by one-way ANOVA]. All these findings indicate that GAP43
and CB1R interact specifically in vitro, which requires GAP43 phos-
phorylation at S41 and inhibits CB1R.

## GAP43 interacts with CB1R in mossy cell axon boutons of the dentate gyrus

To map the CB1R-GAP43 interaction in the mouse brain, we first per-
formed immunofluorescence colocalization assays. Consistent with
previous studies[13,19,20,31,32], we found that GAP43 and CB1R were densely
expressed in the IML of the DG (Supplementary Fig. 1a). To evaluate the
neurochemical identity of the immunolabeled presynaptic boutons,
we used conditional knockout mice in which the CB1R-encoding gene
(*Cnr1*) had been selectively deleted from forebrain GABAergic neurons
(hereafter GABA-*Cnr1*$^{-/-}$ mice)[19] or dorsal telencephalic glutamatergic
neurons (hereafter Glu-*Cnr1*$^{-/-}$ mice)[19]. We found abundant double-
positive puncta for GAP43 and (conceivably glutamatergic-neuron)
CB1R in the IML of GABA-*Cnr1*$^{-/-}$ mice, while colocalization between
GAP43 and (conceivably GABAergic-neuron) CB1R was essentially
absent in Glu-*Cnr1*$^{-/-}$ animals (Supplementary Fig. 1b). The glutama-
tergic boutons of the IML likely correspond to MC axons impinging on
proximal dendrites of GCs[21]. Thus, triple-positive puncta for (con-
ceivably glutamatergic-neuron) CB1R, GAP43, and calretinin (a well-
known MC marker[21]) were visible in high-magnification micrographs of
the IML of GABA-*Cnr1*$^{-/-}$ mice (Supplementary Fig. 1c).

We subsequently conducted PLA experiments in the IML of GABA-
*Cnr1*$^{-/-}$ and Glu-*Cnr1*$^{-/-}$ mice to seek for CB1R-GAP43 complexes. Con-
sistent with our immunostaining data, an overt PLA signal, visualized as
positive puncta, was found in *Cnr1*$^{fl/fl}$ and GABA-*Cnr1*$^{-/-}$ mice, notably
diminishing in Glu-*Cnr1*$^{-/-}$ and full *Cnr1*$^{-/-}$ animals (Fig. 3a). To
unequivocally ascribe CB1R-GAP43 complexes to glutamatergic bou-
tons, we used a Cre-mediated, lineage-specific CB1R genetic rescue
strategy from a *Cnr1*-null background (hereafter Stop-*Cnr1* mice;
Fig. 3b)[33,34]. Thus, we rescued CB1R expression selectively in dorsal
telencephalic glutamatergic neurons (hereafter Glu-*Cnr1*-RS mice) or
forebrain GABAergic neurons (hereafter GABA-*Cnr1*-RS mice). As a
control, a systemic CB1R expression-rescue was conducted (hereafter
*Cnr1*-RS mice). The PLA signal of CB1R-GAP43 complexes was markedly
restored in *Cnr1*-RS and Glu-*Cnr1*-RS mice. In contrast, no significant
rescue of complex expression was observed in GABA-*Cnr1*-RS or Stop-
*Cnr1* animals. Altogether, these findings support that the CB1R-GAP43

interaction occurs at glutamatergic neurons in the IML, presumably on
MC axon boutons.

## Phosphorylated GAP43 inhibits CB1R function at mossy cell to granule cell synapses

CB1Rs at the MC-GC synapse mediate a form of short-term plasticity
known as depolarization-induced suppression of excitatory transmis-
sion (DSE), and tonic suppression of glutamate release[19,24,25].
We therefore examined whether these CB1R-mediated effects
could be impacted by GAP43 interaction. To this end, we generated
AAV1/2 vectors encoding phospho-mimetic GAP43 [AAV1/2-CBA-
GAP43(S41D)-CFP] and phospho-resistant GAP43 [AAV1/2-CBA-
GAP43(S41A)-CFP] fused to the fluorescent reporter CFP. An empty
vector (AAV1/2-CBA-CFP) was used as control. These viral vectors were
injected unilaterally into the hilus of the DG−where MC *somata* are
located−of 3–4-week-old WT mice, and electrophysiological record-
ings were then performed in the contralateral DG to activate com-
missural MC axons expressing the vectors[35]. We confirmed the
presence of CFP-positive fibers selectively in the IML of the con-
tralateral DG (Fig. 4a). Three weeks after viral injection, whole-cell
patch-clamp recordings were performed from GCs, and excitatory
postsynaptic currents (EPSCs) in GCs were evoked by electrical sti-
mulation in the IML. First, we observed that both paired-pulse ratio
[Fig. 4b PPR, Control: 1.16 ± 0.06, $n = 9$ cells; GAP43(S41A): 1.15 ± 0.05,
$n = 8$ cells; GAP43(S41D): 0.92 ± 0.04, $n = 12$ cells; $F_{(2, 26)} = 8.146$; Con-
trol vs GAP43(S41D) $p = 0.0044$, and GAP43(S41A) vs GAP43(S41D)
$p = 0.0087$, by one-way ANOVA] and coefficient of variation
[Fig. 4b CV, Control: 0.29 ± 0.02; GAP43(S41A): 0.30 ± 0.02;
GAP43(S41D): 0.20 ± 0.01; $F_{(2, 26)} = 13.77$; Control vs GAP43(S41D)
$p = 0.0007$, and GAP43(S41A) vs GAP43(S41D) $p = 0.0003$, by one-way
ANOVA] were decreased in GAP43(S41D) compared to control vector
or GAP43(S41A)-injected mice, suggesting that phosphorylation of
GAP43 increases glutamate release probability at MC-GC synapses. To
test this possibility, we applied the CB1R inverse agonist AM251
(5 µM for 10 min) to block constitutively active CB1Rs and measure the
effect on basal transmission. Blocking tonic CB1Rs potentiated MC-GC
transmission in control vector or GAP43(S41A)-injected mice [Fig. 4c,
Control: 134 ± 3.6% of baseline, $n = 5$ cells; $t_{(4)} = 10.70$, $p = 0.0004$ by
two-tailed paired Student's $t$ test; GAP43(S41A): 140 ± 7.9% of baseline,
$n = 5$ cells; $t_{(4)} = 4.67$, $p = 0.0095$ by two-tailed paired Student's $t$ test].
In contrast, AM251 did not induce any potentiation in GAP43(S41D)-
expressing MC-GC synapses [Fig. 4c, GAP43(S41D): 103 ± 2.1% of
baseline, $n = 8$ cells; $t_{(7)} = 0.91$, $p = 0.3911$ by two-tailed paired Student's
$t$ test; $F_{(2, 15)} = 22.80$; Control vs GAP43(S41D) $p = 0.0003$, and
GAP43(S41A) vs GAP43(S41D) $p < 0.0001$, by one-way ANOVA]. Toge-
ther with the reduction in PPR and CV (Fig. 4b), this observation
strongly suggests that the GAP43(S41D)-mediated repression of CB1R
constitutive activity occludes the potentiation mediated by AM251. In
addition, the magnitude of DSE was reduced in GAP43(S41D)-injected
mice compared to control vector-injected or GAP43(S41A)-injected
mice [Fig. 4d, Control: 69.3 ± 1.8% of baseline, $n = 8$ cells; $t_{(7)} = 11.70$,
$p < 0.0001$ by two-tailed paired Student's $t$ test; GAP43(S41A):

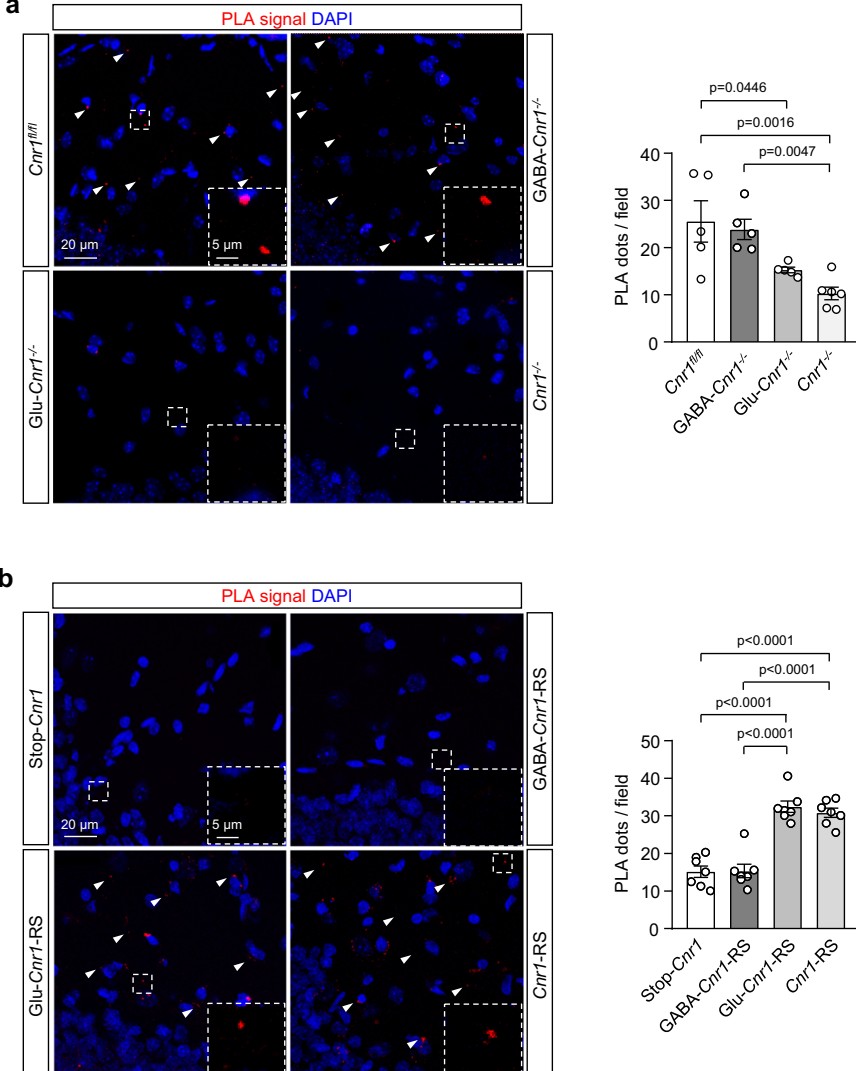

**Fig. 3 | GAP43 interacts with CB1R in MC axon terminals of the DG.** PLA experiments were performed in hippocampal sections from 3-month-old mice of different genotypes. CB$_1$R-GAP43 complexes are shown as PLA-positive red dots. Nuclei are stained with DAPI (blue). **a** Representative images of DG-IML sections from *Cnr1$^{fl/fl}$*, GABA-*Cnr1$^{-/-}$*, Glu-*Cnr1$^{-/-}$*, and full *Cnr1$^{-/-}$* mice. Arrowheads point to some of the complexes. Inset magnifications are included for each genotype. Quantification of PLA-positive dots per field is shown (right, means ± SEM; *Cnr1$^{fl/fl}$* *n* = 5 mice, GABA-*Cnr1$^{-/-}$* *n* = 5 mice, Glu-*Cnr1$^{-/-}$* *n* = 5 mice, *Cnr1$^{-/-}$* *n* = 6 mice; one-way ANOVA with Tukey's multiple comparisons test). **b** Representative images of DG-IML sections from Stop-*Cnr1*, GABA-*Cnr1*-RS, Glu-*Cnr1*-RS, and *Cnr1*-RS mice. Arrowheads point to some of the complexes. Inset magnifications are included for each genotype. Quantification of PLA-positive dots per field is shown (right, means ± SEM; *n* = 7 mice per group; one-way ANOVA with Tukey's multiple comparisons test). Source data are provided as a Source data file.

69.9 ± 4.4% of baseline, *n* = 9 cells; $t_{(8)}$ = 6.91, *p* = 0.0001 by two-tailed paired Student's *t* test; GAP43(S41D): 91.0 ± 2.8% of baseline, *n* = 10 cells; $t_{(9)}$ = 5.38, *p* = 0.0004 by two-tailed paired Student's *t* test; $F_{(2, 24)}$ = 15.48; Control vs GAP43(S41D) *p* = 0.0002, and GAP43(S41A) vs GAP43(S41D) *p* = 0.0002, by one-way ANOVA].

To directly assess CB$_1$R function, we tested the effect of the CB$_1$R agonist WIN-55,212-2 on extracellular MC-GC synaptic responses (i.e., extracellular field excitatory postsynaptic potentials, or fEPSPs) recorded in the IML. While WIN-55,212-2 (5 µM for 25 min) decreased MC-GC fEPSP amplitude in control vector-injected or GAP43(S41A)-injected mice, this effect was attenuated in GAP43(S41D)-injected mice [Fig. 4e, Control: 76.9 ± 3.1% of baseline, *n* = 8 slices; $t_{(7)}$ = 7.59, *p* = 0.0001 by two-tailed paired Student's *t* test; GAP43(S41A): 77.1 ± 3.0% of baseline, *n* = 8 slices; $t_{(7)}$ = 7.83, *p* = 0.0001 by two-tailed paired Student's *t* test; GAP43(S41D): 87.3 ± 2.5% of baseline, *n* = 8 slices; $t_{(7)}$ = 4.85, *p* = 0.0019 by two-tailed paired Student's *t* test; $F_{(2, 21)}$ = 5.695; Control vs GAP43(S41D) *p* = 0.0201, and GAP43(S41A) vs GAP43(S41D) *p* = 0.0225,

by one-way ANOVA]. In contrast, the WIN-55,212-2-mediated reduction of inhibitory synaptic responses (i.e., extracellular field inhibitory postsynaptic potentials, or fIPSPs) recorded in the IML was unaltered under any condition [Fig. 4f, Control: 84.8 ± 2.3% of baseline, *n* = 5 slices; $t_{(4)}$ = 6.06, *p* = 0.0037 by two-tailed paired Student's *t* test; GAP43(S41A): 88.8 ± 2.3 of baseline, *n* = 5 slices; $t_{(4)}$ = 3.98, *p* = 0.0163 by two-tailed paired Student's *t* test; GAP43(S41D): 84.5 ± 2.6% of baseline, *n* = 7 slices; $t_{(6)}$ = 6.11, *p* = 0.0009 by two-tailed paired Student's *t* test; $F_{(2,14)}$ = 0.7654, *p* = 0.4836 by one-way ANOVA]. Taken together, these observations strongly suggest that GAP43, when phosphorylated at S41, inhibits CB$_1$R function at MC-GC synapses.

**GAP43 genetic deletion from mossy cells enhances CB$_1$R synaptic function**

To further characterize the effect of endogenous GAP43 on CB$_1$R function at the MC-GC synapse, we generated *Gap43$^{fl/fl}$* mice (Supplementary Fig. 2a, b, steps i and ii; Supplementary Fig. 2c) and selectively

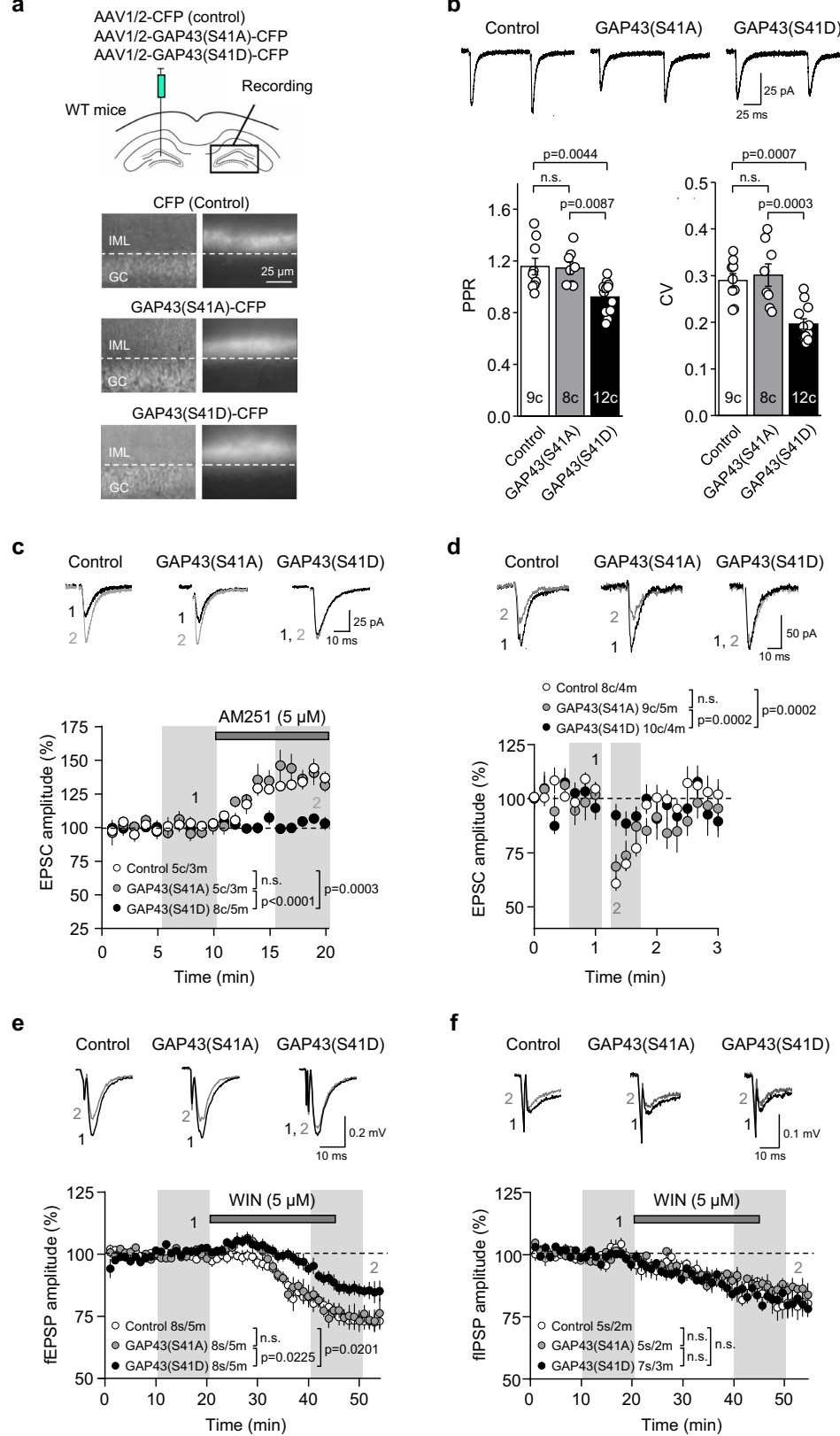

knocked-out *Gap43* from MCs. Briefly, a mix of AAV5-CaMKII-Cre-mCherry and AAV-DG-FLEX-ChIEF-tdTomato was injected in the hilus of *Gap43^{fl/fl}* mice. This manipulation allowed us to selectively and optically stimulate commissural MC axons that lack GAP43 (*Gap43* cKO) and express a fast version of channelrhodopsin (ChIEF). WT mice injected with the same mix of viral vectors were used as control

(Fig. 5a). Cre recombinase activity was confirmed by the reduction of GAP43 labeling at infected MC axon boutons in IML-containing contralateral hippocampal sections (Fig. 5b). Four weeks post-injection, whole-cell patch-clamp recordings were performed from GCs and EPSCs (o-EPSCs) were evoked by optically stimulating MC axons in the IML of the contralateral DG. GAP43-lacking MC-GC synapses displayed

**Fig. 4 | Phosphorylated GAP43 inhibits CB1R function at MC-GC synapses.**
**a** Schematic diagram illustrating the injection of AAV1/2-CBA-CFP (control), AAV1/2-CBA-GAP43(S41A)-CFP or AAV1/2-CBA-GAP43(S41D)-CFP in the hilus of 3-week-old WT mice. Electrophysiological recordings were performed in the contralateral DG. Infrared differential interference contrast (left) and fluorescence (right) images showing CFP expression in the commissural MC axon terminals in the contralateral DG. Note the presence of CFP-positive fibers in the IML and its absence in the GC layer. This CFP-expression pattern was observed in all injected animals with similar results. **b** Whole-cell patch-clamp recordings were performed on GCs from injected mice. Representative traces (top) and quantification bar graph (bottom) for basal PPR and CV are shown (means ± SEM; c = cells; one-way ANOVA with Tukey's multiple comparisons test). **c** EPSCs recorded from injected mice upon AM251 bath application (5 μM, 10 min). Representative EPSC traces, before and after AM251 application (top), and time-course summary plot (bottom) are shown [means ± SEM, c = cells, m = mice; shaded areas indicate the time intervals at which the statistical analysis was conducted; one-way ANOVA with Tukey's multiple

comparisons test]. **d** DSE magnitude in injected mice. Representative traces (top) and time-course summary plot (bottom) are shown [means ± SEM; c = cells, m = mice; shaded areas indicate the time intervals at which the statistical analysis was conducted; one-way ANOVA with Tukey's multiple comparisons test]. **e** fEPSPs recorded from injected mice upon WIN-55,212-2 bath application (WIN; 5 μM, 25 min). Representative fEPSP traces, before and after WIN application (top), and time-course summary plot (bottom) are shown [means ± SEM; s = slices, m = mice; shaded areas indicate the time intervals at which the statistical analysis was conducted; one-way ANOVA with Tukey's multiple comparisons test]. **f** fIPSPs recorded from injected mice upon WIN bath application (5 μM, 25 min). Representative fIPSP traces, before and after WIN application (top) and time-course summary plot (bottom) are shown (means ± SEM; s = slices, m = mice; shaded areas indicate the time intervals at which the statistical analysis was conducted; n.s. by one-way ANOVA with Tukey's multiple comparisons test). Source data are provided as a Source data file.

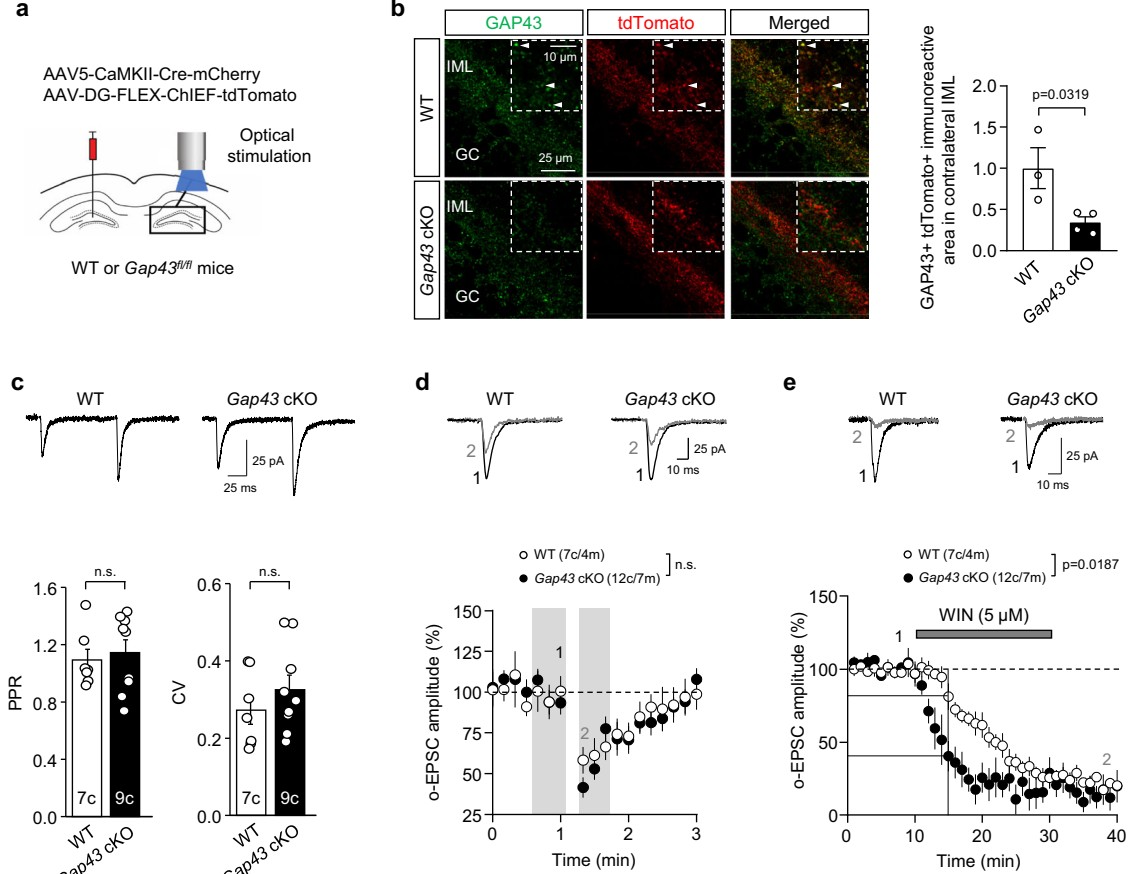

**Fig. 5 | GAP43 genetic deletion from MCs enhances CB1R function at MC-GC synapses.** **a** Schematic diagram illustrating the injection of a mix of AAV5-CamKII-Cre-mCherry and AAV-DG-FLEX-ChIEF-tdTomato in the hilus of 3-week-old *Gap43fl/fl* and WT mice. Light pulses of 0.5–2.0 ms were used to evoke EPSCs driven by the ChIEF-expressing axons originating from commissural MCs. **b** Reduced GAP43 immunoreactivity (green) in AAV-infected tdTomato + (red) fibers of the contralateral IML from injected *Gap43* cKO mice compared to WT mice. Arrowheads point to colocalizing boutons in WT mice. Representative images and quantification of GAP43/tdTomato colocalization are shown (means ± SEM; WT *n* = 3 mice, *Gap43* cKO *n* = 4 mice; two-tailed unpaired Student's *t* test). **c** Whole-cell patch-

clamp recordings of GCs were performed in *Gap43* cKO and WT mice. Representative traces (top) and quantification bar graph for basal PPR and CV (bottom) are shown (means ± SEM; c = cells; n.s. by two-tailed unpaired Student's *t* test). **d** Representative traces (top) and time-course summary plot (bottom) for DSE are shown (means ± SEM; c = cells, m = mice; shaded areas indicate the time intervals at which the statistical analysis was conducted; n.s. by two-tailed unpaired Student's *t* test). **e** Representative traces (top) and time-course summary plot (bottom) for o-EPSC amplitude upon WIN-55,212-2 bath application (WIN; 5 μM, 20 min) (means ± SEM; c = cells, m = mice; WT vs *Gap43* cKO at time15 min, *p* = 0.0187 by two-tailed unpaired Student's *t* test). Source data are provided as a Source data file.

normal PPR (Fig. 5c, WT: 1.09 ± 0.07, *n* = 7 cells; *Gap43* cKO: 1.14 ± 0.09, *n* = 9 cells; $t_{(14)}$ = 0.4248; *p* = 0.6775 by two-tailed unpaired Student's *t* test), CV (Fig. 5c, WT: 0.27 ± 0.04; *Gap43* cKO: 0.33 ± 0.04; $t_{(14)}$ = 0.9890; *p* = 0.3395 by two-tailed unpaired Student's *t* test), and DSE (Fig. 5d; WT: 62.0 ± 7.1% of baseline, *n* = 7 cells; $t_{(6)}$ = 3.42, *p* = 0.0141 by

two-tailed paired Student's *t* test; *Gap43* cKO: 57.4 ± 4.4% of baseline, *n* = 12 cells; $t_{(11)}$ = 7.09, *p* = 0.0001 by two-tailed paired Student's *t* test; WT vs *Gap43* cKO, $t_{(17)}$ = 0.5926, *p* = 0.5612 by two-tailed unpaired Student's *t* test). However, WIN-55,212-2-mediated suppression of MC-GC synaptic transmission was faster in GAP43-deficient compared to

WT MC-GC synapses (Fig. 5e, WT: 81.2 ± 9.0% of baseline, $n = 5$ cells at time 15 min; *Gap43* cKO: 40.3 ± 10.5% of baseline, $n = 5$ cells at time 15 min; $t_{(8)} = 2.941$; $p = 0.0187$ by two-tailed unpaired Student's *t* test), supporting that loss of endogenous GAP43 enhances $CB_1R$ function at MC-GC synapses.

**Enhanced anti-convulsant response to THC in Glu-*Gap43*[−/−] mice**
Next, we aimed to unveil the behavioral relevance of the $CB_1R$-GAP43 interaction. To delete endogenous GAP43 from restricted neuronal subpopulations, we generated conditional knockout mouse lines in which the GAP43-encoding gene was selectively inactivated in dorsal telencephalic glutamatergic neurons or forebrain GABAergic neurons (hereafter Glu-*Gap43*[−/−] and GABA-*Gap43*[−/−] mice, respectively; Supplementary Fig. 2a, b, step iii). Both lines (especially Glu-*Gap43*[−/−]) showed reduced levels of GAP43 protein in the whole hippocampus by Western blot analysis (Supplementary Fig. 2d). This reduction was particularly evident in glutamatergic presynaptic boutons of the IML, as evidenced by immunofluorescence microscopy (Supplementary Fig. 2d). Glu-*Gap43*[−/−] and GABA-*Gap43*[−/−] mice exhibited no noticeable dysmorphology or alteration in survival, growth, and fertility. Both lines, at 8 weeks of age, had normal overall brain morphology (Supplementary Fig. 3a), body weight (Supplementary Fig. 3b), body temperature (Supplementary Fig. 3c), gait pattern (Supplementary Fig. 3d), pain response (Supplementary Fig. 3e), and anxiety-like behavior (Supplementary Fig. 3f). In line with the previously described memory alterations evoked by hippocampal GAP43 inactivation[26,36,37], we found a deficit of long-term novel object recognition memory in Glu-*Gap43*[−/−] mice that was not evident in GABA-*Gap43*[−/−] animals (Supplementary Fig. 3g). Notably, hippocampal $CB_1R$ levels and number of MCs (as identified by calretinin staining) were not affected in any of the two mouse lines (Supplementary Fig. 3h).

Hippocampal MC-GC circuits[22,38], GAP43[39,40], and glutamatergic-neuron $CB_1Rs$[19] have all been implicated in the control of epileptic seizures. Specifically, activation of $CB_1Rs$ with THC and other cannabinoids, under specific therapeutic windows, protects mice[41–43] and other mammals[44,45] against seizures. We therefore evaluated the role of $CB_1R$-GAP43 complexes in this process by using a combined pharmacological and genetic approach. We first tested two doses of the pro-convulsant drug kainic acid (KA; 20 or 30 mg/kg; single i.p. injection), a well-established model of temporal lobe epilepsy[46], to induce excitotoxic epileptic seizures in WT mice (Supplementary Fig. 4a). Administration of THC (10 mg/kg; single i.p. injection) 15 min before KA transiently reduced the mild behavioral seizures induced by 20 mg/kg KA (Supplementary Fig. 4b) but was unable to counteract the more severe pro-epileptic phenotype induced by 30 mg/kg KA (Supplementary Fig. 4c). Hence, we selected the dose of 30 mg/kg KA to be tested in Glu-*Gap43*[−/−] (Fig. 6a) and GABA-*Gap43*[−/−] mice (Supplementary Fig. 5a), aiming to a protective response to THC in the (conceivably $CB_1R$-disinhibited) Glu-*Gap43*[−/−] animals. As expected, no difference in KA-induced seizures over time was evident between vehicle and THC-treated control *Gap43*[fl/fl] mice (Fig. 6b and Supplementary Fig. 5b). In contrast, THC reduced seizure progression in Glu-*Gap43*[−/−] mice (Fig. 6b), while it exerted no protective action in GABA-*Gap43*[−/−] animals (Supplementary Fig. 5b). Consistent with this observation, THC reduced seizure severity (Fig. 6c) and increased latency to seizures (Fig. 6d) in Glu-*Gap43*[−/−] mice but had no effect in GABA-*Gap43*[−/−] animals (Supplementary Fig. 5c, d).

Lastly, to further support the behavioral-seizure data, we monitored electroencephalographic (EEG) activity in the DG following KA administration (30 mg/kg; single i.p. injection). While EEG seizure activity (epilepsy-like spikes; Fig. 6e) was induced in all animals, THC injection increased the latency to tonic-seizure onset (Fig. 6f) and decreased interictal-spike frequency (Fig. 6g) selectively in Glu-*Gap43*[−/−] mice compared to vehicle-treated littermates. No differences in spike duration were observed between the four experimental groups

(Fig. 6h). Collectively, these findings strongly suggest that, upon selective inactivation of GAP43 in glutamatergic neurons, a disinhibited $CB_1R$ reduces the KA-induced pro-convulsant phenotype of mice.

## Discussion
Here, we discovered that GAP43, a protein known for decades for its functions in axonal plasticity, is a $CB_1R$-binding partner. This study also unveils the anatomical and functional $CB_1R$-GAP43 interaction at a particular mouse-brain synapse, thus supporting the notion that $CB_1R$ action can be controlled in a synapse-specific manner. We (present study) and others[27,28] have used MS proteomic-based approaches to define the $CB_1R$ interactome. We were not able to detect various cytoplasmic proteins that have been previously reported to interact with $CB_1R$-CTD, such as CRIP1a, GASP1, SGIP1, FAN and WAVE1 complex, likely owing to differences in the experimental conditions and/or because they were minor components compared to other cellular proteins in our starting brain sample. GAP43 and $CB_1R$ are mainly sorted to presynaptic boutons and anchored in lipid rafts[14,26,30,47], conceivably making them prone to interact. Studies on other potential $CB_1R$ interactors reported to date[27,48–50], including the seminal studies on CRIP1a[51,52], have been mostly conducted with transfected cell lines in culture. Here, we added several molecular techniques to provide a detailed characterization of the interaction of $CB_1R$ with WT and mutant forms of GAP43 in various in vitro and in vivo systems. Phosphorylation of GAP43 by PKC at a unique site (S41) is known to constitute the most relevant modification of GAP43 biological activity[26]. In the present study, we provide robust evidence supporting that the $CB_1R$-GAP43 interaction is dependent on the phosphorylation status of GAP43-S41, in line with previous studies on the interaction of GAP43 with proteins such as SNAP25, syntaxin, F-actin, and rabaptin-5[53–55].

We acknowledge that GAP43(WT) and GAP43(S41D) exhibited a slightly different activity on $CB_1Rs$ depending on the particular protein−protein interaction technique. Specifically, PLA -and, functionally- DMR experiments revealed a similar effect of both constructs on the receptor, while co-immunoprecipitation and BRET experiments suggested that GAP43(S41D) had a stronger effect than GAP43(WT) on the receptor. These observations likely reflect the specific features of each technique. Thus, BRET and PLA detect proteins in very close proximity, the former in a dynamic and highly sensitive fashion in live cells, and the latter as a snapshot on fixed cells requiring a signal-amplification process. In contrast, co-immunoprecipitation may uncover more prominent and stable protein complexes involving direct and indirect protein−protein interactions, while DMR provides an overall agonist-evoked, functional response that conceivably relies on multiple phosphorylation-dephosphorylation cycles and the crosstalk between various signal-transduction cascades. These issues notwithstanding, the phosphorylation-resistant mutant GAP43(S41A) always had a negligible effect on $CB_1R$ irrespective of the technique used, thus supporting the necessity of GAP43 phosphorylation at S41 to enable its binding to $CB_1R$. Regarding the mapping of $CB_1R$ and GAP43 expression, we found that both proteins reside in close proximity in the DG, showing a high abundance at MC axon boutons of the IML and an absence of expression in GCs. This distribution pattern defines a highly restricted, synapse type-selective occurrence of $CB_1R$-GAP43 complexes in the mouse hippocampus. Nonetheless, we cannot rule out that these complexes exist in other brain regions and/or in diseased states.

Previous reports have also used GAP43-S41 phospho-mimetics to investigate the biological role of GAP43. Of note, compared to WT counterparts, mutant mice with a generalized expression of GAP43(S41D) showed enhanced levels of Hebbian LTP as induced in the perforant path (PP) of the DG in vivo[36], as well as at Schaffer collaterals-CA1 synapses, together with an enhanced PPR[56]. In contrast, the expression of GAP43(WT), GAP43(S41A) or a form of GAP43 with a deletion of the entire effector domain did not affect the level of

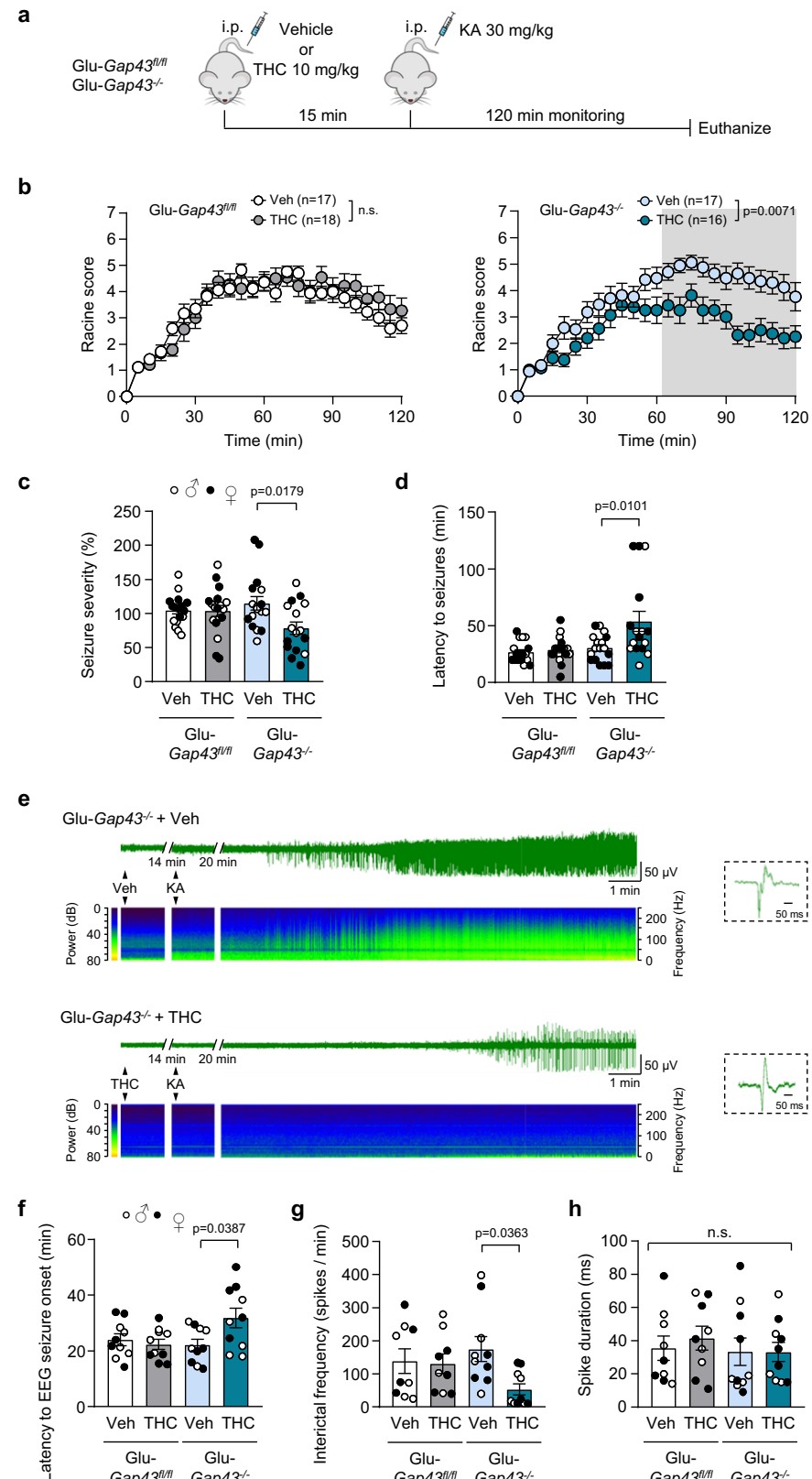

LTP[36,56]. On the other hand, LTD was not affected by GAP43(S41D) overexpression[56], and the induction of LTD in WT animals led to a reduction in GAP43 phosphorylation status[57]. Distinctively, instead of a generalized transgenic model, our approach with AAV-mediated delivery ensures the specific transgene expression at presynaptic boutons of MC axons. Our findings show an inhibitory effect on WIN-

55,212-2-mediated depression of neurotransmission and DSE upon GAP43(S41D) but not GAP43(S41A) expression. The GAP43(S41D)-mediated effects appear remarkably robust despite the coexistence of the GAP43 mutant form with the endogenous protein. The observation that the phospho-resistant GAP43(S41A) construct behaves just like a control empty vector may indicate that, in vivo, the CB$_1$R-GAP43

**Fig. 6 | Enhanced anti-convulsant response to THC in Glu-*Gap43*$^{-/-}$ mice.**
**a** Timeline of the experiments. Vehicle or THC (10 mg/kg, i.p.; 1 injection) was administered to 3-month-old Glu-*Gap43*$^{-/-}$ mice and their corresponding *Gap43*$^{fl/fl}$ littermates. Kainic acid (KA; 30 mg/kg, i.p.; 1 injection) was administered 15 min later, and behavioral score (**b**–**d**) or hippocampal EEG recording (**e**–**h**) was monitored continuously for 120 min. **b** Behavioral scoring of seizures using a modified Racine scale (means ± SEM; number of mice in parentheses; the shaded area indicates all the time points at which $p < 0.05$ by two-way ANOVA with Sidak's multiple comparisons test;). **c** Integrated seizure severity, expressed as normalized percentage from *Gap43*$^{fl/fl}$/Vehicle group (means ± SEM; *Gap43*$^{fl/fl}$/Vehicle $n = 17$ mice, *Gap43*$^{fl/fl}$/THC $n = 18$ mice, Glu-*Gap43*$^{-/-}$/Vehicle $n = 17$ mice, Glu-*Gap43*$^{-/-}$/THC $n = 16$ mice; two-way ANOVA with Tukey's multiple comparisons test). **d** Latency to seizures (means ± SEM; *Gap43*$^{fl/fl}$/Vehicle $n = 17$ mice, *Gap43*$^{fl/fl}$/THC $n = 18$ mice, Glu-

*Gap43*$^{-/-}$/Vehicle $n = 17$ mice, Glu-*Gap43*$^{-/-}$/THC group $n = 16$ mice; two-way ANOVA with Tukey's multiple comparisons test). **e** EEG recordings of representative Glu-*Gap43*$^{-/-}$ mice treated with vehicle (top) or THC (bottom). Epileptic-like spikes appeared after KA injection (insets: detail of individual spikes). The corresponding sonograms (frequency spectrum along recording time) are shown below each recording. **f** Latency to EEG seizure onset (means ± SEM, $n = 10$ mice per group; two-way ANOVA with Tukey's multiple comparisons test). **g** Interictal frequency (means ± SEM, *Gap43*$^{fl/fl}$/Vehicle $n = 9$ mice, *Gap43*$^{fl/fl}$/THC n = 9 mice, Glu-*Gap43*$^{-/-}$/Vehicle $n = 10$ mice, Glu-*Gap43*$^{-/-}$/THC $n = 10$ mice; two-way ANOVA with Tukey's multiple comparisons test). **h** Average spike duration (means ± SEM; *Gap43*$^{fl/fl}$/Vehicle $n = 9$ mice, *Gap43*$^{fl/fl}$/THC $n = 9$ mice, Glu-*Gap43*$^{-/-}$/Vehicle $n = 10$ mice, Glu-*Gap43*$^{-/-}$/THC $n = 10$ mice; n.s. by two-way ANOVA with Tukey's multiple comparisons test). Source data are provided as a Source data file.

interaction does not stand as a constant interaction but rather occurs under specific biological triggers (e.g., those relying on high-activity regimes associated with PKC activation). Therefore, the CB$_1$R-GAP43 interaction would most likely reflect a transient functional state.

The high CB$_1$R expression at MC axon boutons supports a robust negative control of MC-GC synaptic transmission and LTP[25,35]. Remarkably, LTP at PP-GC synapses is associated with increased GAP43 mRNA expression in MCs[58]. Thus, activity-dependent upregulation of GAP43 phosphorylation and levels, and the resultant inhibition of CB$_1$R signaling, could be a mechanism by which information transfer is fine-tuned in the DG, possibly contributing to DG-dependent learning. To complement the GAP43-overexpression approach, we combined Cre-mediated promotor-driven recombination and optogenetics to allow a precise spatiotemporal modulation of MC axon boutons lacking GAP43. We found that the selective deletion of endogenous GAP43 in MCs markedly enhanced the CB$_1$R-mediated inhibition of MC-GC synaptic transmission but had no significant effect on DSE. While phosphorylated GAP43 strongly suppressed MC-GC basal transmission and DSE (Fig. 4), consistent with a reduction in CB$_1$R function, removing GAP43 alone using a conditional knockout approach was not sufficient to strongly disinhibit CB$_1$R function in acute brain slices (Fig. 5). These observations suggest that the CB$_1$R-GAP43 interaction alone may not be sufficient to inhibit CB$_1$R function, and that GAP43 must be phosphorylated to effectively control CB$_1$Rs. In addition, the mild synaptic phenotype observed in our GAP43-deficient MCs could result from compensatory changes (e.g., by other CB$_1$R-modulating proteins) and/or incomplete GAP43 deletion (e.g., due to deficient Cre recombination).

GAP43 has previously been implicated in epileptogenesis[40]. GAP43 expression levels increase at MC axon boutons of the IML upon seizure induction prior to mossy fiber sprouting[39,59]. Moreover, MCs regulate GC activity directly through innervation or indirectly through modulation of GABAergic interneurons[21,22]. MC-evoked excitation of GCs is normally weak, but becomes significantly strengthened following pro-convulsant insults[22,60]. On the other hand, CB$_1$Rs can dampen overactive neural circuits. For example, glutamatergic-neuron CB$_1$R in the DG is crucial for attenuating the KA-induced epileptic phenotype in mice, presumably by stabilizing the recurrent GC-MC-GC circuitry[19]. A specific down-regulation of CB$_1$R protein and mRNA on glutamatergic but not GABAergic axon terminals has been reported in epileptic human hippocampal tissue[61]. In addition, somatic transfer of the CB$_1$R-encoding gene to hippocampal glutamatergic neurons was sufficient to protect mice against acute seizures and neuronal damage[62]. CB$_1$R agonists can act as anti-convulsants in various animal models of hyperexcitability and epilepsy, presumably by decreasing glutamatergic transmission[42,43,63,64]. Here, we used a dose of THC lower than the ED$_{50}$ values of previous studies, which had reported a mild anti-convulsant activity of the drug in various seizure models[41–43]. Remarkably, only by deleting GAP43 selectively from glutamatergic neurons, including MCs, we were able to achieve an unambiguous THC-mediated anti-convulsant effect, as assessed by both behavioral and EEG assays. It has been reported that THC may also induce pro-

convulsant actions at high doses, likely via CB$_1$R-dependent inhibition of GABAergic neurons[15,65], thus highlighting the relevance of the neuron population-specificity of CB$_1$R action. Understanding this specificity of multimodal cannabinoid signaling may be important to gain further insight into the unwanted effects of cannabis abuse, and to design personalized interventions aimed to enhance or depress CB$_1$R activity in selective pathological situations.

## Methods
### Animals
Experimental procedures were performed in accordance with the guidelines and approval of the Animal Welfare Committees of Universidad Complutense de Madrid, Universidad Autónoma de Madrid and Comunidad de Madrid, the directives of the Spanish Government and the European Commission, as well as the guidelines of NIH and Albert Einstein College of Medicine Institutional Animal Care and Use Committee. Throughout the study, animals had unrestricted access to food and water. They were housed (typically, 4–5 mice per cage) under controlled temperature (range, 20–22 °C), humidity (range, 50-55%) and light/dark cycle (12 h/12 h). Animal housing, handling, and assignment to the different experimental groups was conducted by standard procedures[9]. Adequate measures were taken to minimize pain and discomfort of the animals. Both males and females, at approximately 1:1 ratio within each experimental group, were used in this study. We employed C57BL/6J wild-type mice (Charles River) for GAP43-AAV injections and subsequent acute hippocampal slice preparation (Fig. 4; see below). The rest of the experiments were conducted with the following mutant-mouse lines, all of them generated in C57BL/6N background: *Cnr1*$^{floxed/floxed}$ (*Cnr1*$^{fl/fl}$) mice, *Cnr1*$^{floxed/floxed;CMV-Cre}$ (*Cnr1*$^{-/-}$) mice, conditional *Cnr1*$^{floxed/floxed;Nex1-Cre}$ (Glu-*Cnr1*$^{-/-}$) mice, and conditional *Cnr1*$^{floxed/floxed;Dlx5/6-Cre}$ (GABA-*Cnr1*$^{-/-}$) mice, to allow CB$_1$R gene deletion from a WT background[19,66,67]; as well as Stop-*Cnr1* mice, Stop-*Cnr1*$^{Ella-Cre}$ (*Cnr1*-RS) mice, conditional Stop-*Cnr1*$^{Nex1-Cre}$ (Glu-*Cnr1*-RS) mice, and conditional Stop-*Cnr1*$^{Dlx5/6-Cre}$ (GABA-*Cnr1*-RS) mice, to allow CB$_1$R gene-expression rescue from a CB$_1$R-null background[33,34]. As for GAP43 animal models, we purchased B6Dnk;B6Brd;B6N-Tyrc-Brd *Gap43*$^{tm1a(EUCOMM)Wtsi/WtsiBiat}$ mice from the EMMA Mouse Repository (MGI ID #5700649). This mouse line contains a *LacZ* cassette and a *Neo* cassette with a Stop codon flanked by *Frt* sites, followed by the *Gap43* exon 2 flanked by *LoxP* sites. These mice were crossed with mice carrying a constitutive expression of Flp recombinase under the control of the constitutive promoter *ACTB* (The Jackson Laboratory, strain #005703; kindly provided by Dr. Rui Benedito, CNIC, Madrid, Spain), thus allowing a rescued, conditional-ready floxed allele (*Gap43*$^{fl/fl}$). Subsequent crossing with *Nex1*-Cre or *Dlx5/6*-Cre-expressing mice[19] yielded the corresponding conditional knockout mouse lines (Glu-*Gap43*$^{-/-}$ and GABA-*Gap43*$^{-/-}$, respectively). All the generated *Gap43* conditional knockout mice were backcrossed in C57BL/6N background for at least 8 generations before use. The following primers were used for the genotyping of these mice (5′–3′ sequence; F: Forward, R: Reverse): *LacZ*_F ATCACGACGCGCTGTATC, *LacZ*_R

ACATCGGGCAAATAATATCG, *Gap43 Frt*_F TGGACGCTTAGGGGAGAG AG, *Gap43 Frt*_R TTCCAGGGCTCAAGAAAAGG, CAS_R TCGTGGTATCG TTATGCGCC, *Gap43* WT/floxed_F TGACACAAACTGGGGTCAGA, GAP 43 WT_R GGGATGAAAGGCTATTAGATCTGT, *Gap43* floxed_R GGCGAG CTCAGACCATAACT, *Flp*_F CTAATGTTGTGGGAAATTGGAGC, *Flp*_R CTCGAGGATAACTTGTTTATTGC. The main characteristics of all these *Cnr1* and *Gap43* genetically-modified mouse lines, and which experiments they were used in, are shown in Supplementary Table 2.

## Affinity-based proteomics

A whole sheep brain was homogenized by mechanical disaggregation in RIPA buffer (50 mM Tris-HCl, 150 mM NaCl, 1% v/v Triton X-100, 0.1% w/v deoxycholic acid, 0.1% w/v SDS, pH 7.35). The soluble fraction of the homogenate was loaded onto a Sepharose 4B column after extensively washing the column with 100 mM Tris-HCl, 200 mM NaCl, pH 7.0. The eluted soluble fraction was then collected and loaded onto the Sepharose 4B column saturated with purified hCB$_1$R-CTD bound to lectin. Purified lectin-empty plasmid was expressed in an additional Sepharose 4B column as a control. After washing with RIPA and TBS buffer (50 mM Tris-HCl, 150 mM NaCl, pH 7.0), the bound fraction was eluted with 200 mM lactose. Proteins were then subjected to nLC/MS-MS proteomic analysis. Briefly, the samples were loaded on a 12% acrylamide gel and a denaturing electrophoresis was carried out. Once samples had reached the resolving gel, electrophoresis was stopped, and the gel was died with Coomassie Colloidal Blue overnight. After fading, the region of the gel containing the sample was cut just above the recombinant protein, this piece being divided into smaller fractions that were thereafter digested with trypsin. The resulting peptide fragments were retained in an Acclaim PepMap 100 precolumn (Thermo Scientific, Waltham, MA, USA) and then eluted in an Acclaim PepMap 100 C18 column, 25-cm long, 75 μm-internal diameter, and 3-μm particle size (Thermo Scientific). The peptides were separated in a gradient for 110 min (90 min 0–35% of Buffer B; 10 min 45–95% Buffer B; 9 min 95% Buffer B; and 1 min 10% Buffer B; Buffer B supplemented with 0.1% formic acid in acetonitrile) at 250 nL/min in a nanoEasy nLC 1000 (Proxeon) coupled to an ionic source with nanoelectrospray (Thermo Scientific) for electrospray ionization. Mass spectra were acquired in an LTQ–Orbitrap Velos mass spectrometer (Thermo Scientific) working in positive mode, which measures the mass-to-charge ratio (*m/z*) of ionized particles and detects the relative number of ions at each m/z ratio. Mass spectra corresponding to a full screening (*m/z* 400 to 2000) were obtained with a resolution of 500,000–60,000 (*m/z* = 200), and the 15 most intense ions from each screening were selected for fragmentation by cleavage at peptide bonds via collision with a gaseous matrix by collision-induced dissociation with the energy of collision normalized to 35%. Ions with unique charge or no charge were discarded. A dynamic exclusion of 45-s duration was conducted. The masses of the fragments were then determined by ion trap to define the amino acid sequence of the peptides. Proteomic spectrum files were challenged to data bases and Uniprot of sheep (*Ovis aries*) (UPID: UP000002356) and other mammals (*Mammalian*) [taxonomy_id:40674] using Proteome Discoverer software version 1.4.1.14 (Thermo Scientific) and the searching tool SEQUEST (built-in version of Proteome Discoverer version 1.4.1.14). In the searching criteria, carbamide methylation, cysteine nitrosylation and methionine oxidation were established as dynamic modifications. Tolerance to precursor selection and product ions was fixed at 10 ppm and 0.5 Da, respectively. Peptide identification was validated by the Percolator algorithm (built-in version of Proteome Discoverer version 1.4.1.14) using $q \leq 0.01$ (*q*-value being the *p* value additionally adjusted to the False Discovery Rate).

## Acute hippocampal slice preparation

Viral vectors were stereotaxically injected into the hilus of 3–4-week-old C57BL/6J mice. Briefly, mice were anesthetized with isoflurane

(up to 5% for induction and 1–2% for maintenance) and either AAV1/2-CBA-CFP, AAV1/2-CBA-GAP43(S41A)-CFP or AAV1/2-CBA-GAP43(S41D)-CFP were injected into the hilus (1 μL at a flow rate of 0.1 μL/min) using the following coordinates (mm to bregma): antero-posterior −2.18, medio-lateral ±1.5, dorso-ventral −2.2 from dura. Coordinates were adjusted according to the bregma to lambda distance for each mouse. At 3 weeks post-injection, acute hippocampal slices were prepared. Animals were anesthetized with isoflurane, and brains were removed and rapidly transferred into ice-cold cutting solution containing (in mM): 110 choline chloride, 25 NaHCO$_3$, 1.25 NaH$_2$PO$_4$, 2.5 KCl, 0.5 CaCl$_2$, 7 MgCl$_2$, 25 D-glucose, 11.6 sodium L-ascorbate, and 3.1 sodium pyruvate. Hippocampi were isolated and sliced (300-μm thick) using a VT1200S microslicer (Leica Microsystems Co.). Slices were then transferred and incubated for 30 min in a chamber placed at 33–34 °C and with artificial cerebrospinal fluid (ACSF) solution containing (in mM): 124 NaCl, 26 NaHCO$_3$, 10 D-glucose, 2.5 KCl, 1 NaH$_2$PO$_4$, 2.5 CaCl$_2$, and 1.3 MgSO$_4$. Slices were kept at room temperature for at least 45 min prior to experiments. All solutions were maintained at 95% O$_2$/5% CO$_2$ (pH 7.4).

## Electrophysiology

All recordings were performed at $28 \pm 1\,°C$ in a submersion-type recording chamber perfused at ~2 mL/min with ACSF supplemented with GABA$_A$ receptor antagonist picrotoxin (100 μM), except for inhibitory synaptic transmission that was monitored in the presence of D-APV (50 μM) and NBQX (10 μM). Whole-cell patch-clamp recordings were made from GCs voltage clamped at −60 mV using a patch-type pipette electrode (~3–4 MΩ) containing (in mM): 131 Cs-gluconate, 8 NaCl, 1 CaCl$_2$, 10 EGTA, 10 D-glucose, and 10 HEPES, pH 7.2 (285–290 mOsm). Series resistance (~8–28 MΩ) was monitored throughout all experiments with a −5 mV, 80 ms voltage step, and cells that exhibited a change in series resistance (>20%) were excluded from analysis. Extracellular field recordings (fEPSPs and fIPSCs) were performed using a patch-type pipette filled with 1 M NaCl and placed in the IML. All experiments were performed in an interleaved fashion−i.e., control experiments were performed every test experiment on the same day.

The stimulating patch-type pipette was filled with ACSF and placed in the IML (<50 μm from the border of the GC body layer) to activate MC axons. To elicit synaptic responses, paired, monopolar square-wave voltage pulses (100–200 μs pulse width, 4–25 V) were delivered through a stimulus isolator (Isoflex, AMPI) connected to a broken tip (10–20 μm) stimulating patch-type pipette filled with ACSF. Stimulus intensity was adjusted to get comparable magnitude synaptic responses across experiments (e.g., 50–100 pA EPSCs at Vh = −60 mV or 5 mV fEPSPs). Stimulation was achieved by delivering paired pulses 100 ms apart. PPR was defined as the ratio of the amplitude of the second EPSC, to the amplitude of the first EPSC. CV was calculated as the standard deviation of EPSC amplitude divided by the mean of EPSC amplitude. Both PPR and CV were measured during 10-min baseline. In drug-delivery experiments, stimulation was triggered every 20 s, and every 10 s in DSE experiments. DSE was induced with 5-s depolarizing voltage step (from −60 mV to 0 mV) to trigger endocannabinoid release from GCs. The magnitude of DSE was determined as the percentage change between the mean amplitude of 3 consecutive EPSCs preceding DSE and the mean amplitude of 3 consecutive EPSCs following depolarization. For WIN-55,212-2-mediated depression, the magnitude of depression was calculated by comparing the 10-min baseline responses with the 10-min responses at the end of WIN-55,212-2 application (specifically, from 5 min before washout to 5 min after washout). Representative traces were obtained by averaging 15 individual traces.

Electrophysiological data were acquired at 5 kHz, filtered at 2.4 kHz, and analyzed using IgorPro software version 7.01 (Wavemetrics, Inc.)[25]. Picrotoxin, WIN-55,212-2, AM251, D-APV and NBQX

were purchased from Tocris-Cookson Inc. (Ellisville, MO, USA) and dissolved in water or DMSO. Total DMSO in the bath solution was maintained at 0.1% in all experiments.

## Optogenetics

Gap43$^{fl/fl}$ mice were anesthetized with isoflurane (up to 5% for induction and 1–2% for maintenance). A mix of AAV5-CaMKII-Cre-mCherry and AAV-DG-FLEX-ChIEF-tdTomato viruses (ratio 1:2) was stereotaxically injected into the hilus, using the same coordinates as mentioned above. Acute hippocampal slices were prepared as described above 4 weeks after viral injection, and whole-cell patch clamp recordings were performed in the contralateral DG. Pulses of blue light (0.5–2 ms) were provided by using a 473-nm LED (Thorlabs, Inc.) through the microscope objective (×40, 0.8 NA), and centered in the IML.

## Behavioral tests

Adult (ca. 3-month-old) Glu-Gap43$^{-/-}$ and GABA-Gap43$^{-/-}$ mice, as well as their respective Gap43$^{fl/fl}$ control littermates, were used for behavioral tests. Animals were assigned randomly to the different treatment groups, habituated to the experimental room, and handled 1 week before testing. All the behavioral tests were video-recorded for subsequent blind analysis by a different trained observer, using Smart3.0 Software (Panlab, Barcelona, Spain). Body weight and temperature were measured, the latter with a thermo-coupled flexible probe (Panlab, Madrid, Spain) located in the rectum for 10 s. Analgesia was evaluated by the hot-plate paradigm. The test consisted of placing a mouse on an enclosed hot plate (Columbus Instruments, Columbus, OH, USA) and measuring the latency to lick one of the paws. Walking patterns were monitored with ink-painted paws (blue, fore; red, hind) on 70-cm long paper sheets. Anxiety-like behaviors were assessed in the elevated plus-maze test. The maze consisted of a cross-shaped plastic device with two 30-cm long, 5-cm wide opposite open arms, and two 30-cm long, 5-cm wide, 16-cm high opposite closed arms, connected by a central structure (5 × 5 cm) and elevated 50 cm from the floor. First, a 5-min observation session was performed, in which each mouse was placed on the central neutral zone facing one of the open arms. The cumulative time spent in the open and closed arms was then recorded. One arm entry was considered when the animal had placed at least both forelimbs in the arm. Data are represented as the number of visits to the open arms. For the novel object recognition task, mice were habituated for 9 min in an L-maze[68]. Twenty-four hours later, mice were first exposed to two identical objects located at the edges of the maze arms and allowed exploration for 9 min during the training session. After an inter-trial interval of 24 h, mice were placed back again for a 9-min test session in which one of the familiar objects had been replaced by a novel object. Object exploration was defined as the mouse directing its nose to the object (>2 cm) and being involved in active exploration. Mice did not show preference for any object before trials. A discrimination index ($I_d$) was calculated to measure recognition memory as $(t_{new\ object} - t_{(familiar\ object)})/(t_{new\ object} + t_{(familiar\ object)})$, being the denominator the total exploration time. Mice showing total exploration times below 15 s were excluded from the analyses.

For induction of acute excitotoxic seizures, KA (Sigma) was dissolved in isotonic saline, pH 7.4. Seizures were induced by injecting 20 mg/kg or 30 mg/kg of KA i.p. in a volume of 10 mL/kg body weight[19,67]. An i.p. injection of vehicle (1% v/v DMSO in 1:18 v/v Tween-80/saline solution) or 10 mg/kg THC (THC Pharm) was administered 15 min prior to KA injection. Right after, mice were placed in clear plastic cages and recorded and monitored continuously for 120 min− or, occasionally, until death−for seizure behavior. The higher score reached in each 5-min interval was given according to the following modified Racine scale:[69] immobility (stage 1); forelimb and/or tail extension, rigid posture (stage 2); repetitive movements and head bobbing (stage 3); rearing and falling (stage 4); continuous rearing and falling, jumping, and/or wild running (stage 5); generalized tonic-clonic

seizures (stage 6); and death (stage 7)[19]. Integrated seizure severity was determined[70] by the formula Seizure severity = Σ(all scores of a given mouse)/time of experiment. The latency to seizures was assessed as the time from KA injection to reach stage 3 of Racine scale.

## Hippocampal electroencephalograhy (EEG)

Adult (ca. 3-month-old) Glu-Gap43$^{-/-}$ and their Gap43$^{fl/fl}$ control littermates were anesthetized with urethane (1.6 g/kg body weight, i.p.) and placed on the stereotaxic apparatus, in which body temperature was kept at 37 °C throughout the experimental process with a water-heated pad (Gaymar T/Pump, Orchard Park, NY, USA). The sagittal midline of the scalp was sectioned and retracted, and a small craniotomy was drilled over the hippocampus. Hippocampal EEG was recorded in the right brain hemisphere through tungsten macroelectrodes (1 MΩ; A-M Systems, Sequim, WA, USA) that were stereotaxically implanted in the DG at the following coordinates (mm to bregma): antero-posterior −2.4, medio-lateral ±1.5, dorso-ventral −2.0 from dura. Field potentials were filtered in the 0.3–100 Hz interval and amplified using a DAM50 preamplifier (World Precision Instruments, Sarasota, FL, USA). Signals were sampled at 200 Hz through an analog-to-digital converter (Power 1401 data acquisition unit, Cambridge Electronic Design, Cambridge, UK) and analyzed with Spike2 software version 10 (Cambridge Electronic Design).

Each experiment began with 5 min of basal recording, followed by 15 min of recording after treatment with either vehicle or THC (10 mg/kg; single i.p. injection). KA (30 mg/kg; single i.p. injection) was subsequently administered to induce epileptic discharges, and the hippocampal field potential was recorded continuously for the next 120 min− or, occasionally, until death. An epileptic-like spike was defined as a spontaneous potential change of short duration (<100 ms) and a high amplitude (at least twice the standard deviation of the field potential baseline; typically, >20 μV). The latency to seizure onset was defined as the time from KA injection to the beginning of tonic epileptic discharges. The frequency of interictal spikes was measured 60 min after KA injection in a 5-min interval of continuous recording. Spike duration was determined 60 min after KA injection as the average duration of 50 spikes. Power spectra were calculated after vehicle or THC injection and before KA administration using the fast Fourier transform algorithm to characterize hippocampal frequency bands. Owing to anesthesia, these power spectra showed a clear predominance (≥80%) of delta waves (frequency ≤4 Hz) over other hippocampal frequencies, with no overt differences being observed between the different animal genotypes or pharmacological treatments.

## Other methods

The rest of the experimental procedures used in this study are extensively described in Supplementary Methods. That section provides precise details on gene constructs, protein expression and purification, fluorescence polarization, cell culture and transfection, Western blotting and co-immunoprecipitation, synaptosomes preparation, immunofluorescence, PLA, BRET, DMR assays, and recombinant AAV1/2 production. Uncropped scans of all gels and blots are shown in the Source data file (blots of the main figures) and the Supplementary Information (gels and blots of the supplementary figures; Supplementary Figs. 6 and 7, respectively).

## Statistics

Data are presented as mean ± SEM, and the number of experiments is indicated in every case. Statistical analysis was performed with Graph-Pad Prism version 8.0.1 (GraphPad Software, San Diego, CA, USA). All variables were first tested for normality (Kolmogorov−Smirnov's and Shapiro−Wilk's test) and homocedasticity (Levene's test) before analysis. When variables satisfied these conditions, one-way or two-way ANOVA with Tukey's or Sidak's post hoc test, or two-tailed paired or unpaired Student's $t$ test, was used as appropriate and indicated in

every case. We considered *p* values <0.05 as statistically significant. Power analysis was performed with IBM SPSS software version 28 (IBM, Bois-Colombes, France).

### Reporting summary

Further information on research design is available in the Nature Portfolio Reporting Summary linked to this article.

## Data availability

All data generated or analyzed during the study are included in this article, its Supplementary Information, and its Source data file. Proteomic spectrum files were challenged to data bases and Uniprot of sheep (*Ovis aries*; UPID: UP000002356) and other mammals [*Mammalian*; taxonomy_id: 40674] using Proteome Discoverer software version 1.4.1.14 (Thermo Scientific) and the searching tool SEQUEST (built-in version of Proteome Discoverer version 1.4.1.14). Source data are provided with this paper.

## Code availability

We did not use any new, unreported computer code or algorithm in the study.

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

## Acknowledgements

This work was supported by the Spanish *Ministerio de Ciencia e Innovación* (MICINN/FEDER; grants RTI2018-095311-B-I00 and PID2021-125118OB-I00 to M.G., SAF-2017-87629-R to E.I.C. and V.C., PID2020-113938RB-I00 to E.M. and V.C., BFU 2017-83292-R to J.S.-P., and PID2020-119358GB-I00 to D.F.d.S.) and the NIH (grants R01 MH116673, R01 MH125772, R01 NS115543, and R01 NS113600 to P.E.C.). L.B. and G.M. were supported by INSERM. I.B.M and C.C.-I. were supported by contracts from the Spanish *Ministerio de Universidades* (*Formación de Profesorado Universitario* Program, references FPU15/01833 and FPU16/02593, respectively). We are indebted to Rodrigo Barderas-Machado, Alba Hermoso-López, and Lucía Rivera-Endrinal for expert technical assistance.

## Author contributions

I.B.M., C.C.-I., C.B., E.M., G.M., L.B., E.I.C., V.C., I.G.-R., A.N., D.F.d.S., I.R.-C., P.E.C., and M.G. designed research. I.B.M., C.C.-I., C.B., E.M., A.R.-C., C.M.-F., A.M.-C., R.M., N.G.-F., and A.N. performed research. I.B.M., C.C.-I., C.B., E.M., A.R.-C., R.M., J.S.-P., A.N., D.F.d.S., I.R.-C., P.E.C., and M.G. analyzed data. I.B.M. and M.G. drafted the paper. I.B.M., C.C.-I., C.B., P.E.C., and M.G. wrote the final version of the paper. All the authors revised the final version of the paper. As senior members of each collaboration group on the study, J.S.-P., L.B., V.C., A.N., D.F.d.S., P.E.C., and M.G. take responsibility for their respective group's contribution, including preservation of the original data on which the paper is based; verification that the figures and conclusions accurately reflect the data collected, and that manipulations to images are in accordance with Nature journal guidelines; and minimization of obstacles to sharing materials, data, and algorithms through appropriate planning.

## Competing interests

The authors declare no competing interests.

## Inclusion and ethics

The study followed the inclusion and ethics recommendations set out in the *Nature* journal guidelines when designing, executing, and reporting this research.
