## [Peer Review File · Nature Communications]

CONTROL OF A HIPPOCAMPAL RECURRENT EXCITATORY CIRCUIT BY CANNABINOID RECEPTOR-INTERACTING PROTEIN GAP43REVIEWER COMMENTS

Reviewer #1 (Remarks to the Author):

This study provides data supporting a novel role of GAP-43 as a protein that interacts with CB1R resulting in changes in synaptic excitability within an excitatory hippocampal circuit. The implications of this study could provide the field of neuroscience as a whole many new directions, but also provide neurological disorders associated with hyperexcitability, such as seizure disorders for example, a new therapeutic target. I was excited to read this article and enthusiastic about its direction as other work has shown increased expression of GAP-43 in treatment-resistant epilepsy, and yet some patients only seem to respond to cannabinoid therapeutics.

These experiments elegantly show, for the first time, that GAP43 and CB1R interact presynaptically, most likely in glutamatergic neurons. One of the most interesting results in this study is that decreasing GAP-43 activity by blocking its phosphorylation resulted in reduced glutamate release at MC-GC synapses, and more interestingly, appears to be dependent on CB1R activity. Further support of this relationship is shown by the loss of this finding with knocking out GAP-43. Perhaps most impressive, this paper shows that this relationship is involved in seizures in an excitotoxic model of epilepsy.

I am enthusiastic about these results but have some suggestions on how to improve the results, figures and writing to make the paper more applicable to a broader audience.

This is such an exciting new finding and it opens the door to so many new projects in the field of neuroscience. I was thrilled to see an interaction between CB1R and GAP-43. This manuscript is well-written and provides a good rationale for the study, which will help guide readers without a background in the significance of the study and its results.

Below are some suggestions on how to make the paper more applicable to the general neuroscience community and provide stronger evidence of these interesting results.

Major improvements:

The biggest limitation of this paper is that the Kainic Acid induced seizures were done in the absence of EEG. Since this behavioral data is also the most translational and arguably the most exciting, the EEG data would greatly increase the significance of the study. While behavior is a decent measure of epileptic activity, the study could be missing epileptic activity that is subclinical but would be present by EEG (Rusina et al., 2021). It would make the final in vivo and translational data that this paper provides much stronger if there were EEG associated and if the data were analyzed as 1) latency to seizure onset, 2) duration of seizure, 3) frequency of interictal spikes and 4) duration of time spent in each stage of Racine's behavioral scale to show seizure severity. While the study took measure every 5 minutes to give a behavioral score, it could have missed severe stages of the seizure using this sparse sampling method. Instead, it would be great to see % of the time in each stage.

Minor improvements:

Many abbreviations were never described before they were used in the text.

It would make the paper easier for those without a background in biochemistry to understand if each result were first outlined with what the experiment aimed to determine and describe a bit more about the techniques used as well as write out the abbreviations used, like the DMR experiments on page 8 – this was better explained than the experiments previous to it, especially since the results are reported first in this journal.

Both males and females were used in the research, but was there a sex-specific effect? Were males and females analyzed separately before combining the data?

The different transgenic mice used is a bit confusing. Could a table be put into the supplementary data to show the different strains and which experiments they were used in and their purpose?

It's a bit confusing that the experiments were first conducted in sheep brains (Figure 1A). Is this common in this technique? Why weren't mice used? Perhaps this can be justified in the text for those that are not familiar with this technique.

Further explanation would be nice:

- GAP43 phosphorylation enhances CB1R-GAP43 interaction. Phosphorylation from what? Or just GAP43 phosphorylation from PKC?
- Why were telencephalic and forebrain neurons targeted? Moreover, in the model of selectively inactivating GAP-43 in telencephalic glutamatergic neurons, why was GAP-43 reduced in the hippocampus? Are there anatomical projections that would modulate this?
- What is the mechanism of the CB1R agonist WIN-55,212-2?

Suggestions on changes to the text:

- Line 237 is a bit awkward: "We therefore examined whether these CB1R237 mediated effects could be affected by GAP43 interaction" perhaps "affected" can be replaced with "impacted"
- Line 353 – this language is a bit strong. The paper only investigated the hippocampus in depth, but states that the study "unveils the anatomical and functional specificity of a CB1R-interactor complex at the synapse level..." but what if there are also interactions in other brain areas or under diseased states?

Suggestions on changes to the figures:

- Figure 1C – I am confused as to why there is an opposite result in the bands for GAP43 in the IgG column between the cultured and tissue preparations.
- Figure 1D – it makes sense that only a small subset of presynapses would be GAP43+, therefore, would this data be more striking to express it as % of GAP43+ synaptosomes that are CB1R+ vs. GAP43- synaptosomes that are CB1R-?
- Figure 1E – how was this calculated? The bar graph goes over 100%
- Figure 2B – there appear to be a different number of cells in these images. Perhaps better images within more similar regions would be less bias. Was this quantification method published elsewhere? Counting the PLA dots seems a bit subjective/ambiguous, but if it's been validated in the field or was done in a blinded manner this may be suitable.
- Figure 2D – I am surprised that the GAP43(WT) has the same expression as GAP43(S41A) – I would expect it to be slightly higher, but still lower than GAP43(S41D) but not the same. This could be explained further in the text as to why this does or does not make sense.
- Figure 2E – Is there data on this experiment with a vehicle? It would also be nice to have the mechanism of the inhibitor written into the paper briefly. I am surprised that there is such a similar trace for the WT and GAP43(S41D). However, the authors argue in the results that phosphorylation is

important for this finding. If so, I would expect GAP43(S41D) to have a different result than WT.

- There are no statistical analyses for Figure 2 D and E.
- Figure 3 is incredibly difficult to see except for the zoomed in panels. It would be nice to show zoomed in views of all of these conditions. The significance bars on Figure 3B bar graphs appear to be incorrect as there are two of the same bars for the same conditions. Please check this.
- The graphs in Figure 4C are really hard to see. Perhaps the circles can be smaller to make this easier to interpret or different colors can be used.
- Figure 4D does not show statistical significance.
- Figure 5E does not show statistical significance.
- Figure 6B does not show statistical significance.
- Figure S3 A the GABA-Gap43 anterior section is much more anterior than the compared Glu-Gap43 section. Approximate stereotaxic coordinates for these sections would be helpful.

Reviewer #2 (Remarks to the Author):

This is an excellent manuscript by Maroto and colleagues. The authors report that CB1 cannabinoid receptors, one of the most abundant G protein-coupled receptors in the CNS are binding partners of GAP43, a widely distributed presynaptic adaptor protein. Considering the central function of CB1 receptors in the regulation of neurotransmitter release, it is somewhat surprising that the molecular mechanisms that facilitates anchoring the CB1 protein at release sites has remained enigmatic for so long. Therefore, the present finding fills a long-standing gap in our knowledge. Notably, the authors also provide ample evidence about the functional importance of GAP43-CB1 interaction in the regulation of synaptic transmission as well as it's pathophysiological significance in an animal model of temporal lobe epilepsy. The observation that the GAP43-CB1 interaction has a synapses-specific importance within the dentate gyrus circuitry, specifically at the mossy cell-granule cell synapses is especially exciting. Using optostimulation of the commissural axons in the assay is very elegant.

The manuscript is well-written, the experiments are elegant and appropriate, the results are convincing and the discussion is fair and concise.

I have only a few minor comments:

1. Lines 255-257. I agree that the increased glutamate release probability by mimicking GAP43 phosphorylation (S41D) indirectly suggests reduced tonic cannabinoid receptor signaling as the authors propose. However, considering the widespread presynaptic regulatory role of GAP43 as an adaptor protein, other mechanisms may also account for the observed changes in paired-pulse ratio and coefficient of variation. As a more direct experiment, did the authors apply a CB1 antagonist/inverse agonist to determine if there is indeed reduced tonic cannabinoid signaling?
2. While application of the CB1 agonist WIN55,212-2 unmask facilitated CB1 signaling, neither baseline synaptic transmission nor depolarization-induced suppression of excitation is altered in the GAP43 knockout mice. Please discuss in more details why the endogenous endocannabinoid signaling mechanisms (tonic signaling reflected in baseline parameters and phasic signaling reflected in DSE) remain unimpaired in the absence of GAP43.

3. In Fig 6B, is there a hidden stronger epileptic phenotype in Glu-GAP43^{-/-} mice in case of the vehicle treatment indicating a tonic control of GAP43-CB1 interaction in excitatory circuits? What is the p value for a post hoc comparison for these two groups and what is the strength of the data based on power analysis?

Reviewer #3 (Remarks to the Author):

The authors report that the C-terminal, intracellular domain of the cannabinoid CB1 receptor interacts in vivo specifically with the known growth-associated protein GAP-43. This protein acts as a regulatory partner of the CB1 receptor. The main conclusion is that GAP-43 regulates the receptor function.

Phosphorylation of GAP-43 facilitates its interaction with the cannabinoid receptor. The authors also found that the GAP-43- CB1 receptor interaction occurs at glutamatergic neurons and that GAO43 deletion enhances CB1 receptor synaptic function.

These are noteworthy results of major significance to the cannabinoid field. They are completely original and I assume that numerous groups in the research area will now investigate GAP-43. The work supports the conclusions and claims and I did not notice any flaws in the data analysis, interpretation and conclusions.

Numerous methods and large number of results are presented in the Supplementary. In my view some of these data could be the basis of a separate publication.

The methodology is sound, but I have to state that I do not have experience with many of the methods used. I believe that enough details are provided in the methods for the work to be reproduced.

All the research was done with the CB1 cannabinoid receptor. I assume that work with the CB2 receptor will be initiated.

I suggest that the publication be accepted without any changes.

POINT-BY-POINT RESPONSE TO THE REVIEWERS' COMMENTS

Reviewer #1 (Remarks to the Author):

This study provides data supporting a novel role of GAP-43 as a protein that interacts with CB1R resulting in changes in synaptic excitability within an excitatory hippocampal circuit. The implications of this study could provide the field of neuroscience as a whole many new directions, but also provide neurological disorders associated with hyperexcitability, such as seizure disorders for example, a new therapeutic target. I was excited to read this article and enthusiastic about its direction as other work has shown increased expression of GAP-43 in treatment-resistant epilepsy, and yet some patients only seem to respond to cannabinoid therapeutics.

These experiments elegantly show, for the first time, that GAP43 and CB1R interact presynaptically, most likely in glutamatergic neurons. One of the most interesting results in this study is that decreasing GAP-43 activity by blocking its phosphorylation resulted in reduced glutamate release at MC-GC synapses, and more interestingly, appears to be dependent on CB1R activity. Further support of this relationship is shown by the loss of this finding with knocking out GAP-43. Perhaps most impressive, this paper shows that this relationship is involved in seizures in an excitotoxic model of epilepsy.

I am enthusiastic about these results but have some suggestions on how to improve the results, figures and writing to make the paper more applicable to a broader audience.

This is such an exciting new finding and it opens the door to so many new projects in the field of neuroscience. I was thrilled to see an interaction between CB1R and GAP-43. This manuscript is well-written and provides a good rationale for the study, which will help guide readers without a background in the significance of the study and its results.

Below are some suggestions on how to make the paper more applicable to the general neuroscience community and provide stronger evidence of these interesting results.

We would like to thank the reviewer for his/her positive and constructive comments, which have helped to improve the quality of our study.

Major improvements:

The biggest limitation of this paper is that the Kainic Acid induced seizures were done in the absence of EEG. Since this behavioral data is also the most translational and arguably the most exciting, the EEG data would greatly increase the significance of the study. While behavior is a decent measure of epileptic activity, the study could be missing epileptic activity that is subclinical but would be present by EEG (Rusina et al., 2021). It would make the final in vivo and translational data that this paper provides much stronger if there were EEG associated and if the data were analyzed as 1) latency to seizure onset, 2) duration of seizure, 3) frequency of interictal spikes and 4) duration of time spent in each stage of Racine's behavioral scale to show seizure severity. While the study took measure every 5 minutes to give a behavioral score, it could have missed severe stages of the seizure using this sparse sampling method. Instead, it would be great to see % of the time in each stage.

We agree with the reviewer. We have therefore conducted hippocampal EEG experiments and measured various EEG-associated parameters in *Glu-Gap43*^{-/-} mice, in which we had previously found an altered response of behavioral epileptic seizures to kainic acid/THC (see

new Figs 6E-H; and text, lines 382-390 and 697-728). In addition, as indicated by the reviewer, we have re-analyzed our previous behavioral data for seizure severity and latency to seizures -and have added new animals to allow a more reliable statistical evaluation (see new Figs 6B-D and S5B-D; and text, lines 379-381 and 694-695). Taken together, these new EEG and behavioral data provide further support to our previous finding that deleting GAP43 selectively from glutamatergic neurons enhances the antiepileptic activity of CB1R agonism.

Regarding the duration of time spent in each stage of Racine's behavioral scale (item #4 suggested by the reviewer), as the scale comprises many stages (specifically, 7), we frankly believe that we would need a higher number of animals, including males and females for a disaggregated assessment (see below), to allow a biologically and statistically meaningful analysis. In any event, the analysis conducted on our cohort of animals shows a marked difference between vehicle- and THC-treated *Glu-Gap43^{-/-}* mice (with low-damage stages prevailing upon THC treatment), as well as a milder difference between vehicle- and THC-treated *Gap43^{fl/fl}* mice (with a propensity to high-damage stages prevailing upon THC treatment):

Finally, we understand that severe stages were not missed because we visualized the whole video of each animal continuously, not only every 5 min, and then gave the higher score of the Racine's scale reached in each 5-min interval. We apologize for the confusion as this point was not well explained in the text. Now we have clarified it (Methods, subsection "Behavioral tests", lines 686-688).

Minor improvements:

Many abbreviations were never described before they were used in the text.

Sorry for this error. We have double-checked that all non-standard abbreviations are described when cited for the first time in the text.

It would make the paper easier for those without a background in biochemistry to understand if each result were first outlined with what the experiment aimed to determine and describe a bit more about the techniques used as well as write out the abbreviations used, like the DMR experiments on page 8 – this was better explained than the experiments previous to it, especially since the results are reported first in this journal.

Good remark indeed. We have amended the text accordingly (lines 144-145 for fluorescence polarization, 168-170 for PLA, and 189-191 for BRET).

Both males and females were used in the research, but was there a sex-specific effect? Were males and females analyzed separately before combining the data?

Both male and female mice, at approximately 1:1 ratio within each experimental group, were used in the study. Source data were collected and analyzed as disaggregated for sex

(see Source data file). We have now included specific symbols for males and females in each panel where appropriate (Figs 6, S3, S4 and S5). Except -as expected- for body weight (Fig S3B), which was higher in males than in females, no significant sex-specific differences were found in the numerous parameters measured in the study. We are nonetheless aware that trends appeared in a few cases (for example, when assessing seizure severity in *Glu-Gap43*^{-/-} mice; Fig 6C), but they did not reach statistical significance. We cannot rule out, however, that sample size was not high enough to enable meaningful *post hoc* statistical conclusions.

The different transgenic mice used is a bit confusing. Could a table be put into the supplementary data to show the different strains and which experiments they were used in and their purpose?

Great idea. We have elaborated a table with those items (see Supplementary Information, Table S2, cited in the text on lines 534-536).

It's a bit confusing that the experiments were first conducted in sheep brains (Figure 1A). Is this common in this technique? Why weren't mice used? Perhaps this can be justified in the text for those that are not familiar with this technique.

Sheep brain rather than mouse brain was used just to provide a much bigger amount of starting biological material for the large-scale proteomic experiment. This has been explained in the text (lines 126-127).

Further explanation would be nice:

- GAP43 phosphorylation enhances CB1R-GAP43 interaction. Phosphorylation from what? Or just GAP43 phosphorylation from PKC?

Yes, PKC is the kinase that phosphorylates the GAP43-S41 residue. We have specified it in the text (lines 177-178, 216 and 311-312).

Why were telencephalic and forebrain neurons targeted? Moreover, in the model of selectively inactivating GAP-43 in telencephalic glutamatergic neurons, why was GAP-43 reduced in the hippocampus? Are there anatomical projections that would modulate this?

We used Nex1-Cre mice to generate conditional mutant mice lacking GAP43 or CB1R expression selectively in glutamatergic neurons from the telencephalon because these two proteins exhibit their most prominent expression and function in this part of the brain. Likewise, we used in parallel Dlx5/6-Cre mice to generate conditional mutant mice lacking GAP43 or CB1R expression selectively in GABAergic neurons from the forebrain. We have employed throughout the manuscript the term "telencephalon" in a broad sense, as the Nex1 promoter is active in neurons not only from the neocortex but also from the paleocortex (see Soria-Gómez et al. Nat Neurosci 2014, 17:407-415, for a detailed characterization of CB1R deletion from glutamatergic neurons of the olfactory circuitry in Nex1-CB1R KO mice) and the archicortex (see Monory et al. Neuron 2006, 51:455-466, for a detailed characterization of CB1R deletion from glutamatergic neurons of the hippocampus, especially mossy cells, in Nex1-CB1R KO mice). Hence, we got the sought Nex1-Cre-driven deletion of either GAP43 or CB1R from mossy cells in our respective Nex1-conditional KOs. We have clarified this issue in Table S2. We are unaware of any anatomical projections that can indirectly affect Nex1-Cre-mediated genetic recombination in the hippocampus.

- What is the mechanism of the CB1R agonist WIN-55,212-2?

WIN-55,212-2 is a well-established synthetic cannabinoid receptor full agonist with high affinity, efficacy, and potency on CB1R. It is therefore very widely used and accepted in the cannabinoid-research field -e.g., over 2,000 publications can be found on this compound in PubMed. A brief mention about the main features of WIN-55,212-2 is now included in the text (lines 206-207).

Suggestions on changes to the text:

- Line 237 is a bit awkward: “We therefore examined whether these CB1R237 mediated effects could be affected by GAP43 interaction” perhaps “affected” can be replaced with “impacted”

Done (line 255).

- Line 353 – this language is a bit strong. The paper only investigated the hippocampus in depth, but states that the study “unveils the anatomical and functional specificity of a CB1R-interactor complex at the synapse level...” but what if there are also interactions in other brain areas or under diseased states?

We agree (lines 396-397 and 436-437).

Suggestions on changes to the figures:

- Figure 1C – I am confused as to why there is an opposite result in the bands for GAP43 in the IgG column between the cultured and tissue preparations.

We understand that the IgG column (control condition of both types of preparations) just shows a non-specific smear but not a GAP43-specific band. We have amended the figure by pointing the specific bands with arrowheads. (See also uncropped blots in the Source data file.)

- Figure 1D – it makes sense that only a small subset of presynapses would be GAP43+, therefore, would this data be more striking to express it as % of GAP43+ synaptosomes that are CB1R+ vs. GAP43- synaptosomes that are CB1R-?

We agree with the reviewer. As previously mentioned in the text, only a small subset of total hippocampal synaptosomes (Fig 1D) or neurons (Fig S1A) express GAP43. We have now reanalyzed the data to quantify the number of synaptosomes (*i.e.*, particles stained with the marker synaptophysin 1) that are immunoreactive for either CB1R or GAP43 within the GAP43-positive pool or the CB1R-positive pool, respectively. We found that, in either case, approximately 40% of the bulk hippocampal synaptosomal preparations co-express both proteins. These data have been included in the text (lines 159-165).

- Figure 1E – how was this calculated? The bar graph goes over 100%

Each y-axis value was calculated as the mean of the PLA-positive immunofluorescence signal* of each synaptosome in the counted fields divided by the total number of synaptosomes in those fields (as detected by bright field microscopy). A total of 3 *Cnr1*^{-/-} and 3 WT synaptosomal preparations (each preparation obtained by pooling the hippocampi of 5-6 mice) were used. Then, the mean of the 3 WT preparations was set at 100%. This calculation procedure is now explained in the Supplementary Methods (“Synaptosomal preparations” subsection).

***Please note that synaptosomes are much smaller than whole cells and, therefore, no sufficient resolution is achieved by immunomicroscopy to count individual PLA-positive *puncta* in synaptosomal preparations.**

- Figure 2B – there appear to be a different number of cells in these images. Perhaps better images within more similar regions would be less bias. Was this quantification method published elsewhere? Counting the PLA dots seems a bit subjective/ambiguous, but if it's been validated in the field or was done in a blinded manner this may be suitable.

We agree with the reviewer and have included new images with similar numbers of cells. Yes, this method to quantify PLA-positive *puncta* (indicating that the two proteins of interest are within close proximity) is well validated in the field (*e.g.*, Viñals et al. PLoS Biol 2015, 13:e1002194; Moreno et al. Neuropsychopharmacology 2018, 43:964-977; Blasco-Benito et al. PNAS 2019, 116:3863-3872). We were indeed consistent in the PLA qualifications throughout the study, as the images were always acquired with the same confocal settings for every preparation; counting was always conducted in a blinded manner and applying the same cutting threshold to every image; etc.

- Figure 2D – I am surprised that the GAP43(WT) has the same expression as GAP43(S41A) – I would expect it to be slightly higher, but still lower than GAP43(S41D) but not the same. This could be explained further in the text as to why this does or does not make sense.

We thank the reviewer for raising this concern. Nonetheless, we respectfully ask if he/she is referring to the colP data shown in Fig 2C, in which, indeed, colP-ed GAP43(WT) = colP-ed GAP43(S41A) < colP-ed GAP43(S41D). In any case, regarding GAP43 protein expression, we have not noticed gross differences in the extent of expression in any of the GAP43 WT or mutant constructs by our HEK293T cells used in either PLA, colP, BRET or DMR experiments. This aligns with a previous study that expressed GAP43 S41A/D mutants in primary neurons (Fig 4G in Wang et al. Mol Cell Biol 2015, 35:1712-1726). Likewise, other authors obtained GAP43 WT, S41A and S41D forms from insect cells, and did not find any overt difference in protein stability (Nakamura et al. J Neurochem, 70:983-992). Thus, despite deeply influencing GAP43 function, we believe that the GAP43 S41A/D mutations do not affect GAP43 protein expression in our experimental setting. On the other hand, we note that GAP43(WT) and GAP43(S41D) [though, importantly, not the phosphorylation-resistant mutant GAP43(S41A)] exhibited a slightly different relative activity on CB1R depending on the specific protein-protein interaction technique used, most likely reflecting the unique features of each technique. We have discussed this in the text (lines 416-431).

- Figure 2E – Is there data on this experiment with a vehicle? It would also be nice to have the mechanism of the inhibitor written into the paper briefly. I am surprised that there is such a similar trace for the WT and GAP43(S41D). However, the authors argue in the results that phosphorylation is important for this finding. If so, I would expect GAP43(S41D) to have a different result than WT.

We thank the reviewer for this remark and apologize for not clarifying this issue in the previous version of the manuscript. Each DMR experiment does include a vehicle condition, which routinely gives a negligible-background signal line. Data are obtained by subtracting the corresponding vehicle-datum point from each experimental value. We have now included this information in the “Dynamic mass redistribution (DMR) assay” subsection of Supplementary Methods. As an additional control, a baseline optical signature was always

recorded for 10 min before adding the test compound dissolved in assay buffer with vehicle (0.1% DMSO), which allows a neat stabilization of the signal before cell stimulation. The whole DMR-data analysis followed recommended guidelines in the field (*e.g.*, Schröder et al. Nat Protoc 2011, 6:1748-1760) and, accordingly, we and others have used it in the past to study the signaling evoked by CB1R and other receptors (*e.g.*, Moreno et al. Front Pharmacol 2018, 9:106; Patt et al. J Biol Chem 2021, 296:100472; Costas-Insua et al. J Neurosci 2021, 41:7924-7941). On the other hand, the mechanism by which GAP43(WT) and GAP43(S41D) - but not GAP43(S41A)- inhibit CB1R conceivably relies on a direct physical interaction with the receptor, and so we have explained it in the text (lines 208-210). Finally, as mentioned in the previous point, we are aware that GAP43 exhibited a slightly different relative activity on CB1R depending on the technique used, including DMR, most likely reflecting the unique features of each technique. Of note, however, the phosphorylation-resistant mutant GAP43(S41A) was always ineffective, irrespective of the technique used, thus supporting the necessity of GAP43 phosphorylation at S41 to enable its binding to CB1R. We have discussed this issue in the text (lines 416-431).

- There are no statistical analyses for Figure 2 D and E.

The reviewer is correct. We have now included the statistical analyses for Figs 2D and 2E using the best-established parameter for each technique. For BRET assays, we calculated and analyzed the BRET₅₀ values for each experiment [Results, lines 194-198; and Supplementary Methods, “Bioluminescence resonance energy transfer (BRET)” subsection]. For DMR assays, we calculated and analyzed the DMR_{max} values for each experiment [Results, lines 210-214; and Supplementary Methods, “Dynamic mass redistribution (DMR) assays” subsection]. These data further support the notion that CB1R interacts with and is inhibited by GAP43(S41D) and GAP43(WT), while the phosphorylation-resistant GAP43(S41A) mutant has no effect on the receptor.

- Figure 3 is incredibly difficult to see except for the zoomed in panels. It would be nice to show zoomed in views of all of these conditions. The significance bars on Figure 3B bar graphs appear to be incorrect as there are two of the same bars for the same conditions. Please check this.

As proposed by the reviewer, we have included zoomed pics in all the panels. Yes, the significance bars on Fig 3B were indeed incorrect. We are very sorry for this mistake, which has been amended.

- The graphs in Figure 4C are really hard to see. Perhaps the circles can be smaller to make this easier to interpret or different colors can be used.

The figure has been modified according to the reviewer’s suggestions.

- Figure 4D does not show statistical significance.

The statistical significance has been included.

- Figure 5E does not show statistical significance.

The statistical significance has been included.

- Figure 6B does not show statistical significance.

The statistical significance has been included (Please note that part of former Fig 6, specifically the GABA-Gap43^{-/-} mouse data, has now been moved to new Fig S5.)

• Figure S3 A the GABA-Gap43 anterior section is much more anterior than the compared Glu-Gap43 section. Approximate stereotaxic coordinates for these sections would be helpful.

The reviewer is right. We have changed that section and have amended the figure overall.

Reviewer #2 (Remarks to the Author):

This is an excellent manuscript by Maroto and colleagues. The authors report that CB1 cannabinoid receptors, one of the most abundant G protein-coupled receptors in the CNS are binding partners of GAP43, a widely distributed presynaptic adaptor protein. Considering the central function of CB1 receptors in the regulation of neurotransmitter release, it is somewhat surprising that the molecular mechanisms that facilitates anchoring the CB1 protein at release sites has remained enigmatic for so long. Therefore, the present finding fills a long-standing gap in our knowledge. Notably, the authors also provide ample evidence about the functional importance of GAP43-CB1 interaction in the regulation of synaptic transmission as well as its pathophysiological significance in an animal model of temporal lobe epilepsy. The observation that the GAP43-CB1 interaction has a synapses-specific importance within the dentate gyrus circuitry, specifically at the mossy cell-granule cell synapses is especially exciting. Using optostimulation of the commissural axons in the assay is very elegant.

The manuscript is well-written, the experiments are elegant and appropriate, the results are convincing and the discussion is fair and concise.

We would like to thank the reviewer for his/her positive remarks about our study and for his/her constructive comments.

I have only a few minor comments:

1. Lines 255-257. I agree that the increased glutamate release probability by mimicking GAP43 phosphorylation (S41D) indirectly suggests reduced tonic cannabinoid receptor signaling as the authors propose. However, considering the widespread presynaptic regulatory role of GAP43 as an adaptor protein, other mechanisms may also account for the observed changes in paired-pulse ratio and coefficient of variation. As a more direct experiment, did the authors apply a CB1 antagonist/inverse agonist to determine if there is indeed reduced tonic cannabinoid signaling?

We agree with the reviewer. In response, we have directly tested the effect of the CB1R inverse agonist AM251 on basal transmission to reveal tonic/constitutive CB₁R activity. In line with a previous report (Jensen et al. PNAS 2021, 118:e2017590118), we now show that bath application of AM251 (5 μM for 10 min) increased MC-GC synaptic transmission in CFP- and GAP43(S41A)-injected mice, consistent with the presence of constitutively active CB₁Rs. Conversely, we found that AM251 did not affect basal transmission in GAP43(S41D)-injected mice, suggesting that the effect on CB₁R tonic activity was occluded. Altogether, these findings indicate that the active form of GAP43 [*i.e.*, GAP43(S41D)] inhibits both tonic and phasic activity of CB₁Rs at MC-GC synapses. We report these data in new Fig 4C and the text (lines 274-286).

2. While application of the CB1 agonist WIN55,212-2 unmasks facilitated CB1 signaling, neither baseline synaptic transmission nor depolarization-induced suppression of excitation is altered in the GAP43 knockout mice. Please discuss in more details why the endogenous endocannabinoid signaling mechanisms (tonic signaling reflected in baseline parameters and phasic signaling reflected in DSE) remain unimpaired in the absence of GAP43.

Our results using GAP43 phospho-mimetics strongly indicate that the CB₁R-GAP43 interaction is likely transient and regulated by activity-dependent phosphorylation (e.g., via PKC). While phosphorylated GAP43 strongly suppressed MC-GC basal transmission and DSE, consistent with a reduction in CB₁R function, removing GAP43 using a cKO strategy was insufficient to strongly disinhibit CB₁R function in acute brain slices. This observation suggests that the CB₁R-GAP43 interaction alone may not be sufficient to inhibit CB₁R function and that GAP43 must be phosphorylated to effectively control CB₁Rs. In addition, we do not expect that our cKO strategy will fully delete GAP43 from MCs, and we cannot exclude compensatory mechanisms by other modulatory interacting proteins. The Discussion now includes these points (lines 452-457 and 468-477).

3. In Fig 6B, is there a hidden stronger epileptic phenotype in Glu-GAP43^{-/-} mice in case of the vehicle treatment indicating a tonic control of GAP43-CB1 interaction in excitatory circuits? What is the p value for a post hoc comparison for these two groups and what is the strength of the data based on power analysis?

Glu-GAP43^{-/-} and GAP43^{fl/fl} mice indeed seem to behave somewhat differently under basal, vehicle-treatment conditions at the late time points measured (Fig 6B). Thus, we have conducted additional seizure experiments (reaching n = 17 GAP43^{fl/fl}-vehicle mice and n = 17 Glu-GAP43^{-/-}-vehicle mice) to allow a more reliable statistical evaluation. When analyzing the data by two-way ANOVA, no significant difference came out between the two groups:

ANOVA	SS	DF	MS	F (DFn, DFd)	p value	Power
Genotype	19.88	1	19.88	F (1, 32) = 0.8218	0.3714	0.9999

Likewise, no significant differences between the two groups were found when assessing seizure severity (Fig 6C; p = 0.8159; power = 0.9999) and latency to seizures (Fig 6D; p = 0.9404; power = 1).

Reviewer #3 (Remarks to the Author):

The authors report that the C-terminal, intracellular domain of the cannabinoid CB1 receptor interacts in vivo specifically with the known growth-associated protein GAP-43. This protein acts as a regulatory partner of the CB1 receptor. The main conclusion is that GAP-43 regulates the receptor function.

Phosphorylation of GAP-43 facilitates its interaction with the cannabinoid receptor. The authors also found that the GAP-43- CB1 receptor interaction occurs at glutamatergic neurons and that GAO43 deletion enhances CB1 receptor synaptic function.

These are noteworthy results of major significance to the cannabinoid field. They are completely original and I assume that numerous groups in the research area will now investigate GAP-43.

The work supports the conclusions and claims and I did not notice any flaws in the data analysis, interpretation and conclusions.

Numerous methods and large number of results are presented in the Supplementary. In my view some of these data could be the basis of a separate publication.

The methodology is sound, but I have to state that I do not have experience with many of the methods used. I believe that enough details are provided in the methods for the work to be reproduced.

All the research was done with the CB1 cannabinoid receptor. I assume that work with the CB2 receptor will be initiated.

I suggest that the publication be accepted without any changes.

We would like to thank the reviewer for his/her very positive reaction to our study.

REVIEWERS' COMMENTS

Reviewer #1 (Remarks to the Author):

Thank you for spending so much time addressing my minor concerns. The written text has been clarified for those who are not familiar with the methodology and the results have been further elaborated to help people outside of the field interpret the meaning and impact of these findings. I am truly excited to see this paper come out and I think the work that you have done to improve the paper make it even clearer and stronger in its impact on the field. The changes also make it more applicable to a broader audience that would read Nature Communications. Excellent job on this study and I am glad to have been able to participate in the review.

Reviewer #2 (Remarks to the Author):

In response to the comments of the reviewers, the authors performed new experiments and added more detailed discussion about the implications of their findings. These revisions further strengthened this outstanding paper.

POINT-BY-POINT RESPONSE TO THE REVIEWERS' COMMENTS

Reviewer #1 (Remarks to the Author):

Thank you for spending so much time addressing my minor concerns. The written text has been clarified for those who are not familiar with the methodology and the results have been further elaborated to help people outside of the field interpret the meaning and impact of these findings. I am truly excited to see this paper come out and I think the work that you have done to improve the paper make it even clearer and stronger in its impact on the field. The changes also make it more applicable to a broader audience that would read Nature Communications. Excellent job on this study and I am glad to have been able to participate in the review.

We would like to thank the reviewer very much indeed for his/her very positive reaction to our study.

Reviewer #2 (Remarks to the Author):

In response to the comments of the reviewers, the authors performed new experiments and added more detailed discussion about the implications of their findings. These revisions further strengthened this outstanding paper.

We would like to thank the reviewer very much indeed for his/her very positive reaction to our study.